# A high-performance speech neuroprosthesis

Francis R. Willett[1,15] ✉, Erin M. Kunz[2,3,15], Chaofei Fan[4,15], Donald T. Avansino[1], Guy H. Wilson[5], Eun Young Choi[6], Foram Kamdar[6], Matthew F. Glasser[7,8], Leigh R. Hochberg[9,10,11], Shaul Druckmann[12], Krishna V. Shenoy[1,2,3,12,13,14] & Jaimie M. Henderson[3,6]

Speech brain–computer interfaces (BCIs) have the potential to restore rapid communication to people with paralysis by decoding neural activity evoked by attempted speech into text[1,2] or sound[3,4]. Early demonstrations, although promising, have not yet achieved accuracies sufficiently high for communication of unconstrained sentences from a large vocabulary[1–7]. Here we demonstrate a speech-to-text BCI that records spiking activity from intracortical microelectrode arrays. Enabled by these high-resolution recordings, our study participant—who can no longer speak intelligibly owing to amyotrophic lateral sclerosis—achieved a 9.1% word error rate on a 50-word vocabulary (2.7 times fewer errors than the previous state-of-the-art speech BCI[2]) and a 23.8% word error rate on a 125,000-word vocabulary (the first successful demonstration, to our knowledge, of large-vocabulary decoding). Our participant's attempted speech was decoded at 62 words per minute, which is 3.4 times as fast as the previous record[8] and begins to approach the speed of natural conversation (160 words per minute[9]). Finally, we highlight two aspects of the neural code for speech that are encouraging for speech BCIs: spatially intermixed tuning to speech articulators that makes accurate decoding possible from only a small region of cortex, and a detailed articulatory representation of phonemes that persists years after paralysis. These results show a feasible path forward for restoring rapid communication to people with paralysis who can no longer speak.

It is not yet known how orofacial movement and speech production are organized in motor cortex at single-neuron resolution. To investigate this, we recorded neural activity from four microelectrode arrays—two in area 6v (ventral premotor cortex)[10] and two in area 44 (part of Broca's area)—while our study participant in the BrainGate2 pilot clinical trial attempted to make individual orofacial movements, speak single phonemes or speak single words in response to cues shown on a computer monitor (Fig. 1a,b; Extended Data Fig. 1 shows recorded spike waveforms). Implant locations for the arrays were chosen using the Human Connectome Project multimodal cortical parcellation procedure[10] (Extended Data Fig. 2). Our participant (T12) has bulbar-onset amyotrophic lateral sclerosis (ALS) and retains some limited orofacial movement and an ability to vocalize, but is unable to produce intelligible speech.

We found strong tuning to all tested categories of movement in area 6v (Fig. 1c shows an example electrode). Neural activity in 6v was highly separable between movements: using a simple naive Bayes classifier applied to 1 s of neural population activity for each trial, we could decode from among 33 orofacial movements with 92% accuracy, 39 phonemes with 62% accuracy and 50 words with 94% accuracy (Fig. 1d and Extended Data Fig. 3). By contrast, although area 44 has previously been implicated in high-order aspects of speech production[11–14]

it appeared to contain little to no information about orofacial movements, phonemes or words (classification accuracy below 12%; Fig. 1d). The absence of production-related neural activity in area 44 is consistent with some recent work questioning the traditional role of Broca's area in speech[15–18].

Next, we examined how information about each movement category was distributed across area 6v. We found that speech could be more accurately decoded from the ventral array, especially during the instructed delay period (Fig. 1e), whereas the dorsal array contained more information about orofacial movements. This result is consistent with resting-state functional magnetic resonance imaging (fMRI) data from the Human Connectome Project[10] and from T12 that situates the ventral region of 6v as part of a language-related network (Extended Data Fig. 2). Nevertheless, both 6v arrays contained rich information about all movement categories. Finally, we found that tuning to speech articulators (jaw, larynx, lips or tongue) was intermixed at the single-electrode level (Fig. 1f and Extended Data Fig. 4) and that all speech articulators were clearly represented within both 3.2 × 3.2 mm² arrays. Although previous work using electrocorticographic grids has suggested that there may be a broader somatotopic organization[19] along precentral gyrus, these results suggest that speech articulators are highly intermixed at a single-neuron level.

[1]Howard Hughes Medical Institute at Stanford University, Stanford, CA, USA. [2]Department of Electrical Engineering, Stanford University, Stanford, CA, USA. [3]Wu Tsai Neurosciences Institute, Stanford University, Stanford, CA, USA. [4]Department of Computer Science, Stanford University, Stanford, CA, USA. [5]Department of Neuroscience, Stanford University, Stanford, CA, USA. [6]Department of Neurosurgery, Stanford University, Stanford, CA, USA. [7]Department of Neuroscience, Washington University in St. Louis, St. Louis, MO, USA. [8]Department of Radiology, Washington University in St. Louis, St. Louis, MO, USA. [9]VA RR&D Center for Neurorestoration and Neurotechnology, Rehabilitation R&D Service, Providence VA Medical Center, Providence, RI, USA. [10]School of Engineering and Carney Institute for Brain Science, Brown University, Providence, RI, USA. [11]Center for Neurotechnology and Neurorecovery, Department of Neurology, Massachusetts General Hospital, Harvard Medical School, Boston, MA, USA. [12]Department of Neurobiology, Stanford University, Stanford, CA, USA. [13]Department of Bioengineering, Stanford University, Stanford, CA, USA. [14]Bio-X Program, Stanford University, Stanford, CA, USA. [15]These authors contributed equally: Francis R. Willett, Erin M. Kunz, Chaofei Fan. ✉e-mail: willett2@gmail.com

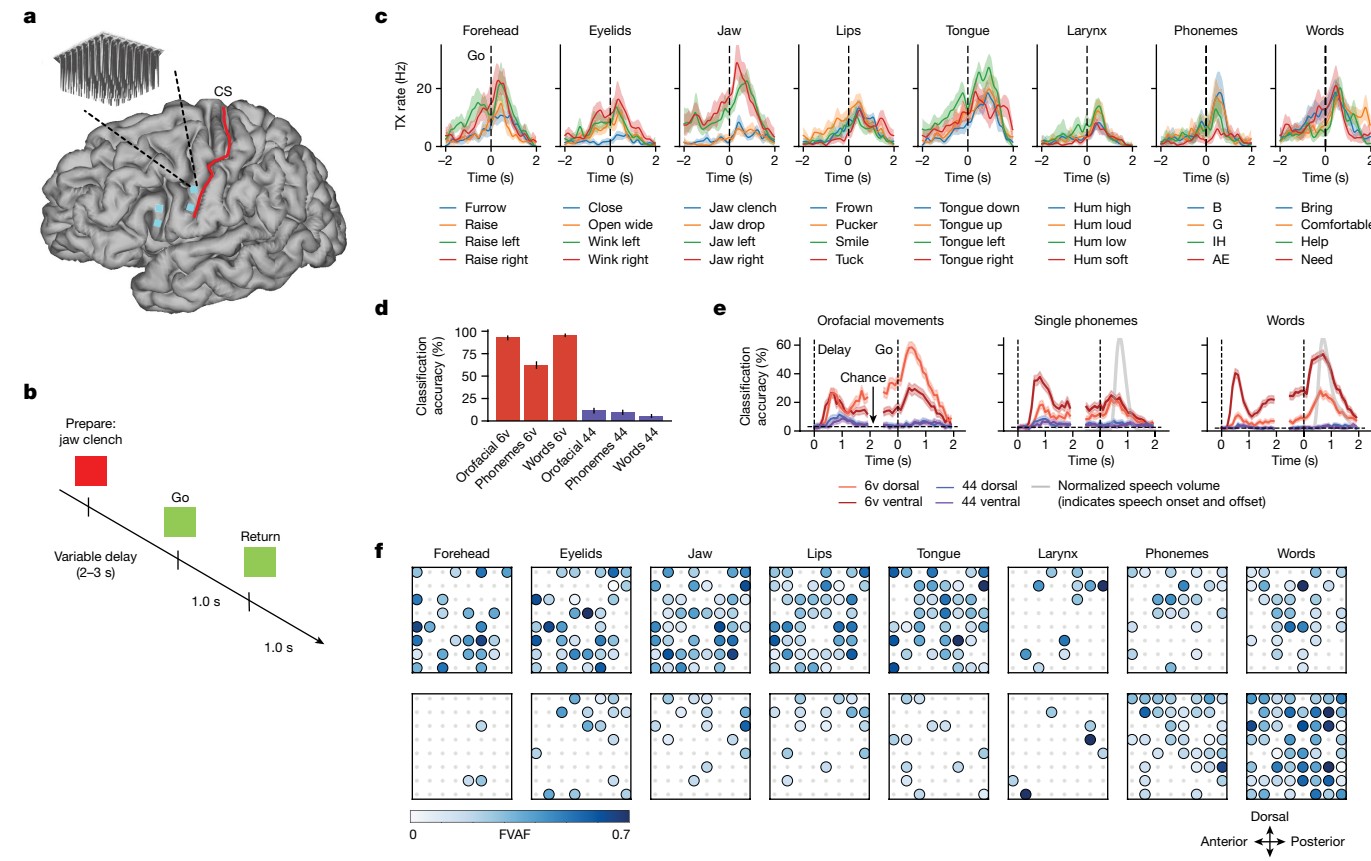

**Fig. 1 | Neural representation of orofacial movement and attempted speech.**
**a**, Microelectrode array locations (cyan squares) are shown on top of
MRI-derived brain anatomy (CS, central sulcus). **b**, Neural tuning to orofacial
movements, phonemes and words was evaluated in an instructed delay task.
**c**, Example responses of an electrode in area 6v that was tuned to a variety of
speech articulator motions, phonemes and words. Each line shows the mean
threshold crossing (TX) rate across all trials of a single condition ($n = 20$ trials
for orofacial movements and words, $n = 16$ for phonemes). Shaded regions
show 95% confidence intervals (CIs). Neural activity was denoised by convolving
with a Gaussian smoothing kernel (80 ms s.d.). **d**, Bar heights denote the
classification accuracy of a naive Bayes decoder applied to 1 s of neural
population activity from area 6v (red bars) or area 44 (purple bars) across all
movement conditions (33 orofacial movements, 39 phonemes, 50 words).
Black lines denote 95% CIs. **e**, Red and blue lines represent classification
accuracy across time for each of the four arrays and three types of movement.
Classification was performed with a 100 ms window of neural population
activity for each time point. Shaded regions show 95% CIs. Grey lines denote
normalized speech volume for phonemes and words (indicating speech onset
and offset). **f**, Tuning heatmaps for both arrays in area 6v, for each movement
category. Circles are drawn if binned firing rates on that electrode were
significantly different across the given set of conditions ($P < 1 \times 10^{-5}$ assessed
with one-way analysis of variance; bin width, 800 ms). Shading indicates the
fraction of variance accounted for (FVAF) by across-condition differences in
mean firing rate.

In sum, robust and spatially intermixed tuning to all tested move-
ments suggests that the representation of speech articulation is prob-
ably sufficiently strong to support a speech BCI, despite paralysis and
narrow coverage of the cortical surface. Because area 44 appeared to
contain little information about speech production, all further analyses
were based on area 6v recordings only.

## Decoding attempted speech

Next, we tested whether we could neurally decode whole sentences
in real time. We trained a recurrent neural network (RNN) decoder to
emit, at each 80 ms time step, the probability of each phoneme being
spoken at that time. These probabilities were then combined with a
language model to infer the most probable underlying sequence of
words, given both the phoneme probabilities and the statistics of the
English language (Fig. 2a).

At the beginning of each RNN performance-evaluation day we first
recorded training data during which T12 attempted to speak 260–
480 sentences at her own pace ($41 \pm 3.7$ min of data; sentences were
chosen randomly from the switchboard corpus[20] of spoken English).
A computer monitor cued T12 when to begin speaking and what

sentence to speak. The RNN was then trained on these data in com-
bination with all previous days' data, using custom machine learning
methods adapted from modern speech recognition[21–23] to achieve high
performance on limited amounts of neural data. In particular, we used
unique input layers for each day to account for across-day changes in
neural activity, and rolling feature adaptation to account for within-day
changes (Extended Data Fig. 5 highlights the effect of these and other
architecture choices). By the final day our training dataset consisted
of 10,850 total sentences. Data collection and RNN training lasted for
140 min per day on average (including breaks).

After training, the RNN was evaluated in real time on held-out sen-
tences that were never duplicated in the training set. For each sentence,
T12 first prepared to speak the sentence during an instructed delay
period. When the 'go' cue was given, neural decoding was automatically
triggered to begin. As T12 attempted to speak, neurally decoded words
appeared on the screen in real time reflecting the language model's
current best guess (Supplementary Video 1). When T12 had finished
speaking she pressed a button to finalize the decoded output. We
used two different language models: a large-vocabulary model with
125,000 words (suitable for general English) and a small-vocabulary
model with 50 words (suitable for expressing some simple sentences

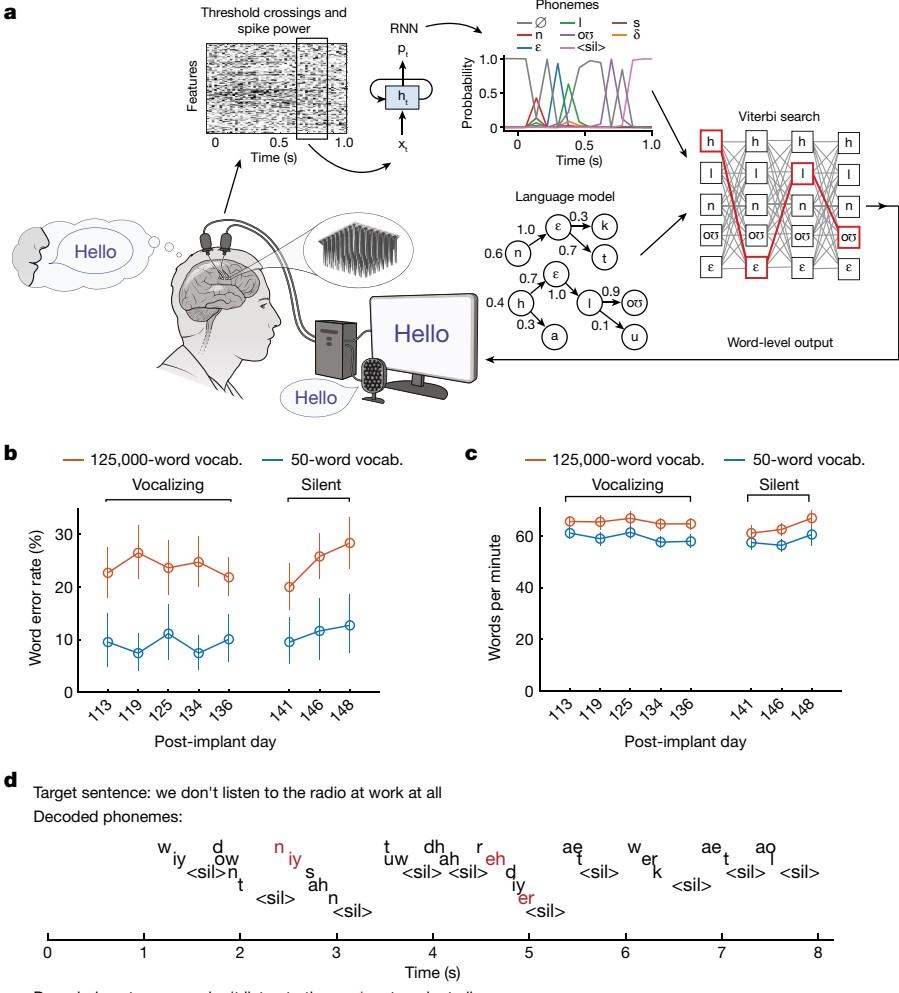

**Fig. 2 | Neural decoding of attempted speech in real time. a**, Diagram of the decoding algorithm. First, neural activity (multiunit threshold crossings and spike power) is temporally binned and smoothed on each electrode. Second, an RNN converts a time series of this neural activity into a time series of probabilities for each phoneme (plus the probability of an interword 'silence' token and a 'blank' token associated with the connectionist temporal classification training procedure). The RNN is a five-layer, gated recurrent-unit architecture trained using TensorFlow 2. Finally, phoneme probabilities are combined with a large-vocabulary language model (a custom, 125,000-word trigram model implemented in Kaldi) to decode the most probable sentence. Phonemes in this diagram are denoted using the International Phonetic Alphabet. **b**, Open circles denote word error rates for two speaking modes

(vocalized versus silent) and vocabulary size (50 versus 125,000 words). Word error rates were aggregated across 80 trials per day for the 125,000-word vocabulary and 50 trials per day for the 50-word vocabulary. Vertical lines indicate 95% CIs. **c**, Same as in **b**, but for speaking rate (words per minute). **d**, A closed-loop example trial demonstrating the ability of the RNN to decode sensible sequences of phonemes (represented in ARPABET notation) without a language model. Phonemes are offset vertically for readability, and '<sil>' indicates the silence token (which the RNN was trained to produce at the end of all words). The phoneme sequence was generated by taking the maximum-probability phonemes at each time step. Note that phoneme decoding errors are often corrected by the language model, which still infers the correct word. Incorrectly decoded phonemes and words are denoted in red.

useful in daily life). Sentences from the switchboard corpus[20] were used to evaluate the RNN with the 125,000-word vocabulary. For the 50-word vocabulary we used the word set and test sentences from Moses et al.[2].

Performance was evaluated over 5 days of attempted speaking with vocalization and 3 days of attempted silent speech ('mouthing' the words with no vocalization, which T12 reported she preferred because it was less tiring). Performance was consistently high for both speaking modes (Fig. 2b,c and Table 1). T12 achieved a 9.1% word error rate for the 50-word vocabulary across all vocalizing days (11.2% for silent) and a 23.8% word error rate for the 125,000-word vocabulary across all vocalizing days (24.7% for silent). To our knowledge, this is the first successful demonstration of large-vocabulary decoding and is also a significant advance in accuracy for small vocabularies (2.7 times fewer errors than in a previous work[2]). These accuracies were achieved at high speeds: T12 spoke at an average pace of 62 words per minute, which

more than triples the speed of the previous state of the art for any type of BCI (18 words per minute for a handwriting BCI[8]).

Encouragingly, the RNN often decoded sensible sequences of phonemes before a language model was applied (Fig. 2d). Phoneme error rates computed on the raw RNN output were 19.7% for vocal speech (20.9% for silent; see Table 1) and phoneme decoding errors followed a pattern related to speech articulation, in which phonemes that are articulated similarly were more likely to be confused by the RNN decoder (Extended Data Fig. 6). These results suggest that good decoding performance is not overly reliant on a language model.

We also examined how information about speech production was distributed across the electrode arrays (Extended Data Fig. 7). We found that, consistent with Fig. 1, the ventral 6v array appeared to contribute more to decoding. Nevertheless, both arrays were useful and low word error rates could be achieved only by combining both (offline analyses

**Table 1 | Mean phoneme and word error rates (with 95% CIs) for the speech BCI across all evaluation days**

| | Phoneme error rate, % (95% CI) | Word error rate, % (95% CI) |
|---|---|---|
| **Online** | | |
| 125,000-word, vocal | 19.7 (18.6, 20.9) | 23.8 (21.8, 25.9) |
| 125,000-word, silent | 20.9 (19.3, 22.6) | 24.7 (22.0, 27.4) |
| 50-word, vocal | 21.4 (19.6, 23.2) | 9.1 (7.2, 11.2) |
| 50-word, silent | 22.1 (19.9, 24.3) | 11.2 (8.3, 14.4) |
| **Offline** | | |
| 125,000-word, improved LM | 19.7 (18.6, 20.9) | 17.4 (15.4, 19.5) |
| 125,000-word, improved LM + proximal test set | 17.0 (15.7, 18.3) | 11.8 (9.8, 13.9) |

Phoneme error rates assess the quality of the RNN decoder's output before a language model is applied, whereas word error rates assess the quality of the combined RNN and language model (LM) pipeline. CIs were computed with the bootstrap percentile method (resampling over trials 10,000 times). Online refers to what was decoded in real time whereas offline refers to post hoc analysis of data using an improved language model (improved LM) or different partitioning of training and testing data (proximal test set). In the proximal test set, training sentences occur much closer in time to testing sentences, mitigating the effect of within-day neural non-stationarities.

showed a reduction in word error rate from 32 to 21% when adding the dorsal to the ventral array).

Finally we explored the ceiling of decoding performance offline by (1) making further improvements to the language model and (2) evaluating the decoder on test sentences that occurred closer in time to the training sentences (to mitigate the effects of within-day changes in the neural features across time). We found that an improved language model could decrease word error rates from 23.8 to 17.4%, and that testing on more proximal sentences further decreased word error rates to 11.8% (Table 1). These results indicate that substantial gains in performance are probably still possible with further language model improvements and more robust decoding algorithms that generalize better to

non-stationary data (for example, unsupervised methods that track non-stationarities without the requirement for new training data[24–27]).

## Preserved representation of speech

Next we investigated the representation of phonemes in area 6v during attempted speech. This is a challenging problem because we do not have ground-truth knowledge of when each phoneme is being spoken (because T12 cannot speak intelligibly). To estimate how each phoneme was neurally represented, we analysed our RNN decoders to extract vectors of neural activity ('saliency' vectors) that maximized RNN probability output for each phoneme. We then asked whether these saliency vectors encode details about how phonemes are articulated.

First we compared the neural representation of consonants to their articulatory representation, as measured by electromagnetic articulography in an able-bodied speaker. We found a broadly similar structure, which is especially apparent when ordering consonants by place of articulation (Fig. 3a); the correlation between electromagnetic articulography (EMA) and neural data was 0.61, far above chance (Fig. 3b). More detailed structure can also be seen—for example, nasal consonants are correlated (M, N and NG)—and W is correlated with both labial consonants and velar/palatal consonants (because it contains aspects of both). Examining a low-dimensional representation of the geometry of both neural and articulatory representation shows a close match in the top two dimensions (Fig. 3c).

Next we examined the representation of vowels, which have a two-dimensional articulatory structure: a high versus low axis (height of the tongue in the mouth, corresponding to the first formant frequency) and a front versus back axis (whether the tongue is bunched up towards the front or back of the mouth, corresponding to the second formant frequency). We found that the saliency vectors for vowels mirror this structure, with vowels that are articulated similarly having a similar neural representation (Fig. 3d,e). Additionally, neural activity contains a plane that reflects the two dimensions of vowels in a direct way (Fig. 3f).

Finally we verified these results using additional ways of estimating neural and articulatory structure and with additional able-bodied speakers (Extended Data Fig. 8). Taken together, these results show

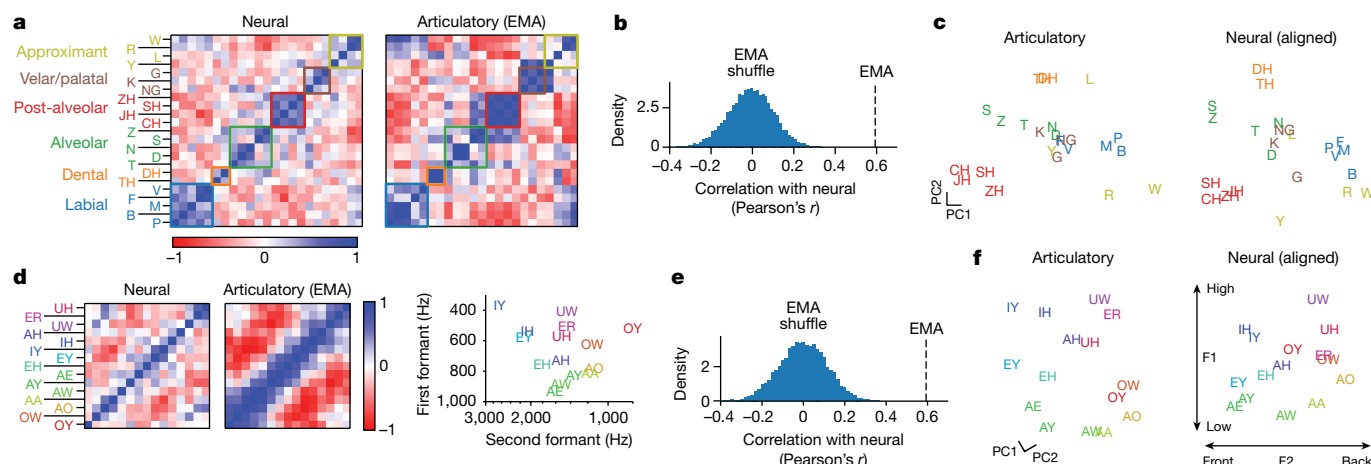

**Fig. 3 | Preserved articulatory representation of phonemes.**
**a**, Representational similarity across consonants for neural data (left) and articulatory data from an example subject who can speak normally, obtained from the USC-TIMIT database (right). Each square in the matrix represents pairwise similarity for two consonants (as measured by cosine angle between neural or articulatory vectors). Ordering consonants by place of articulation shows a block-diagonal structure in neural data that is also reflected in articulatory data. **b**, Neural activity is significantly more correlated with an articulatory representation than would be expected by chance. The blue distribution shows correlations expected by chance (estimated from 10,000 reshufflings of phoneme labels). **c**, Low-dimensional representation

of phonemes articulatorily (left) and neurally (right). Neural data were rotated within the top eight principal components (PC), using cross-validated Procrustes, to show visual alignment with articulatory data. **d**, Representational similarity for vowels, ordered by articulatory similarity. Diagonal banding in the neural similarity matrix indicates a similar neural representation. For reference, the first and second formants of each vowel are plotted below the similarity matrices[38]. **e**, Neural activity correlates with the known two-dimensional structure of vowels. **f**, Same as **c** but for vowels, with an additional within-plane rotation applied to align the (high versus low) and (front versus back) axes along the vertical and horizontal.

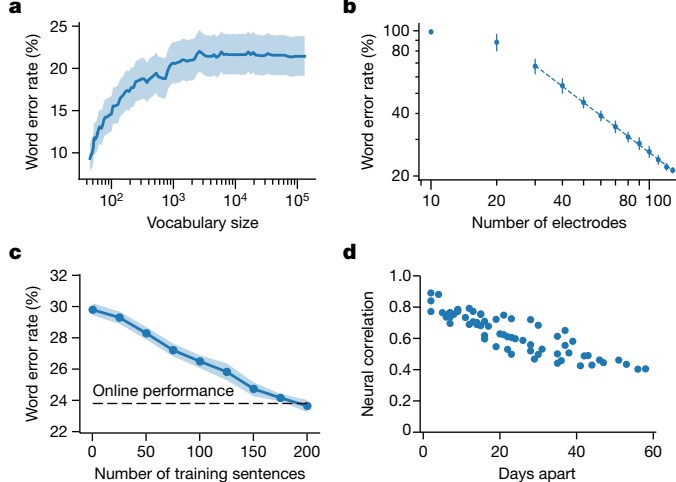

**Fig. 4 | Design considerations for speech BCIs. a**, Word error rate as a function of language model vocabulary size, obtained by reprocessing the 50-word-set RNN outputs with language models of increasingly large vocabulary size. Word error rates were aggregated over the 250 available trials (50 for each of the five evaluation days). The shaded region indicates 95% CI (computed by bootstrap resampling across trials, $n = 10,000$ resamplings). **b**, Word error rate as a function of the number of electrodes included in an offline decoding analysis (each filled circle represents the average word error rate of RNNs trained with that number of electrodes, and each thin line shows s.d. across ten RNNs). There appears to be a log-linear relationship between the number of electrodes and performance, such that doubling the electrode count cuts word error rate by nearly half (factor of 0.57; dashed line represents the log-linear relationship fit with least squares). **c**, Evaluation data from the five vocalized speech-evaluation days were reprocessed offline using RNNs trained in the same way, but with fewer (or no) training sentences taken from the day on which performance was evaluated. Word error rates averaged across ten RNN seeds (blue line) are reasonable even when no training sentences are used from evaluation day (that is, when training on previous days' data only). The shaded region shows 95% CI across the ten RNN seeds (bootstrap resampling method, $n = 10,000$ resamplings). The dashed line represents online performance for reference (23.8% word error rate). **d**, The correlation (Pearson $r$) in neural activity patterns representing a diagnostic set of words is plotted for each pair of days, showing high correlations for nearby days.

that a detailed articulatory code for phonemes is still preserved even years after paralysis.

## Design considerations for speech BCIs

Finally we examined three design considerations for improving the accuracy and usability of speech BCIs: language model vocabulary size, microelectrode count and training dataset size.

To understand the effect of vocabulary size we reanalysed the 50-word-set data by reprocessing the RNN output using language models of increasingly larger vocabulary size (Fig. 4a). We found that only very small vocabularies (for example, 50–100 words) retained the large improvement in accuracy relative to a large-vocabulary model. Word error rates saturated at around 1,000 words, suggesting that use of an intermediate vocabulary size may not be a viable strategy for increasing accuracy.

Next we investigated how accuracy improved as a function of the number of electrodes used for RNN decoding. Accuracy improved monotonically with a log-linear trend (Fig. 4b; doubling the electrode account appears to cut the error rate nearly in half). This suggests that intracortical devices capable of recording from more electrodes (for example, denser or more extensive microelectrode arrays) may be able to achieve improved accuracies in the future, although the extent to which this downward trend will continue remains to be seen.

Finally, in this demonstration we used a large amount of training data per day (260–440 sentences). Retraining the decoder each day helps the decoder to adapt to neural changes that occur across time. We examined offline whether this amount of data per day was necessary by reprocessing the data with RNNs trained with fewer sentences. We found that performance was good even without using any training data on the new day (Fig. 4c; word error rate was 30% with no retraining). Furthermore, we found that neural activity changed at a gradual rate over time, suggesting that unsupervised algorithms for updating decoders to neural changes should be feasible[24–27] (Fig. 4d).

## Discussion

People with neurological disorders such as brainstem stroke or ALS frequently face severe speech and motor impairment and, in some cases, complete loss of the ability to speak (locked-in syndrome[28]). Recently, BCIs based on hand movement activity have enabled typing speeds of between eight and 18 words per minute in people with paralysis[8,29]. Speech BCIs have the potential to restore natural communication at a much faster rate but have not yet achieved high accuracies on large vocabularies (that is, unconstrained communication of any sentence the user may want to say)[1–7]. Here we demonstrate a speech BCI that can decode unconstrained sentences from a large vocabulary at a speed of 62 words per minute, using microelectrode arrays to record neural activity at single-neuron resolution. To our knowledge, this is the first time a BCI has substantially exceeded the communication rates that can be provided by alternative technologies for people with paralysis (for example, eye tracking[30]).

Our demonstration is a proof of concept that decoding attempted speaking movements with a large vocabulary is possible using neural spiking activity. However, it is important to note that it does not yet constitute a complete, clinically viable system. Work remains to be done to reduce the time needed to train the decoder and adapt to changes in neural activity that occur across several days without requiring the user to pause and recalibrate the BCI (see refs. 24–27,31 for initial promising approaches). In addition, intracortical microelectrode array technology is still maturing[32,33] and is expected to require further demonstrations of longevity and efficacy before widespread clinical adoption (although recent safety data are encouraging[34] and next-generation recording devices are under development[35,36]). Furthermore, the decoding results shown here must be confirmed in additional participants, and their generalizability to people with more profound orofacial weakness remains an open question. Variability in brain anatomy is also a potential concern, and more work must be done to confirm that regions of precentral gyrus containing speech information can be reliably targeted.

Importantly, a 24% word error rate is probably not yet sufficiently low for everyday use (for example, compared with a 4–5% word error rate for state-of-the-art speech-to-text systems[23,37]). Nevertheless, we believe that our results are promising. First, word error rate decreases as more channels are added, suggesting that intracortical technologies that record more channels may enable lower word error rates in the future. Second, scope still remains for optimization of the decoding algorithm; with further language model improvements and, when mitigating the effect of within-day non-stationarities, we were able to reduce word error rate to 11.8% in offline analyses. Finally we showed that ventral premotor cortex (area 6v) contains a rich, intermixed representation of speech articulators even within a small area ($3.2 \times 3.2$ mm²), and that the details of how phonemes are articulated are still faithfully represented even years after paralysis in someone who can no longer speak intelligibly. Taken together, these findings suggest that a higher channel count system that records from only a small area of 6v is a feasible path towards the development of a device that can restore communication at conversational speeds to people with paralysis.

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

## Reporting summary

Further information on research design is available in the Nature Portfolio Reporting Summary linked to this article.

## Data availability

All neural data needed to reproduce the findings in this study are publicly available on Dryad (https://doi.org/10.5061/dryad.x69p8czpq). The dataset contains neural activity recorded during the attempted speaking of 10,850 sentences, as well as instructed delay experiments designed to investigate the neural representation of orofacial movement and speech production. As part of this study we also analysed publicly available electromagnetic articulography data: the USC-TIMIT database (https://sail.usc.edu/span/usc-timit/) and the Haskins Production Rate Comparison database (https://yale.app.box.com/s/cfn8hj2puveo65fq54rp1ml2mk7moj3h).

## Code availability

Code that implements an offline reproduction of the central findings in this study (high-performance neural decoding with an RNN) are publicly available on GitHub at https://github.com/fwillett/speechBCI.

**Acknowledgements** We thank participant T12 and her caregivers for their generously volunteered time and effort as part of the BrainGate2 pilot clinical trial; B. Davis, K. Tsou and S. Kosasih for administrative support; Y. Hu for providing suggestions about the language model; and T. Coalson, D. Van Essen and B. Choi for help with the Human Connectome Project pipeline. Support was provided by the Office of Research and Development, Rehabilitation R&D Service, Department of Veterans Affairs (nos. N2864C and A2295R), Wu Tsai Neurosciences Institute, Howard Hughes Medical Institute, Larry and Pamela Garlick, Simons Foundation Collaboration on the Global Brain and NIDCD (nos. U01-DC017844 and U01-DC019430; algorithm development only).

**Author contributions** F.R.W. led the investigation and wrote the initial RNN decoder and task software. C.F. implemented the language models and finalized RNN software. F.R.W., E.M.K. and C.F. optimized and troubleshot the real-time decoding pipeline. F.R.W. and E.M.K. analysed the effect of RNN architecture choices and investigated the representation of speech in 6v and 44. D.T.A. was responsible for rig software architecture and hardware. G.H.W. computed saliency vectors for the articulatory representation analysis. E.Y.C. led the collection of MRI scans and applied the Human Connectome Project cortical parcellation procedure that was used to select array location targets, with the assistance of M.F.G. F.R.W., E.M.K., C.F. and F.K. conducted all other data-collection sessions. F.K. was responsible for coordination of session scheduling, logistics and daily equipment setup/disconnection. L.R.H. is the sponsor-investigator of the multisite clinical trial. J.M.H. planned and performed T12's array placement surgery and was responsible for all clinical trial-related activities at Stanford. F.R.W. wrote the manuscript with the help of E.M.K. and C.F. All authors reviewed and edited the manuscript. The study was supervised and guided by S.D., K.V.S. and J.M.H.

**Competing interests** The MGH Translational Research Center has a clinical research support agreement with Neuralink, Axoft, Reach Neuro and Synchron, for which L.R.H. provides consultative input. J.M.H. is a consultant for Neuralink, serves on the Medical Advisory Board of Enspire DBS and is a shareholder in Maplight Therapeutics. K.V.S. consults for Neuralink and CTRL-Labs (part of Facebook Reality Labs) and is on the scientific advisory boards of MIND-X, Inscopix and Heal. The remaining authors declare no competing interests.

**Additional information**
**Correspondence and requests for materials** should be addressed to Francis R. Willett.

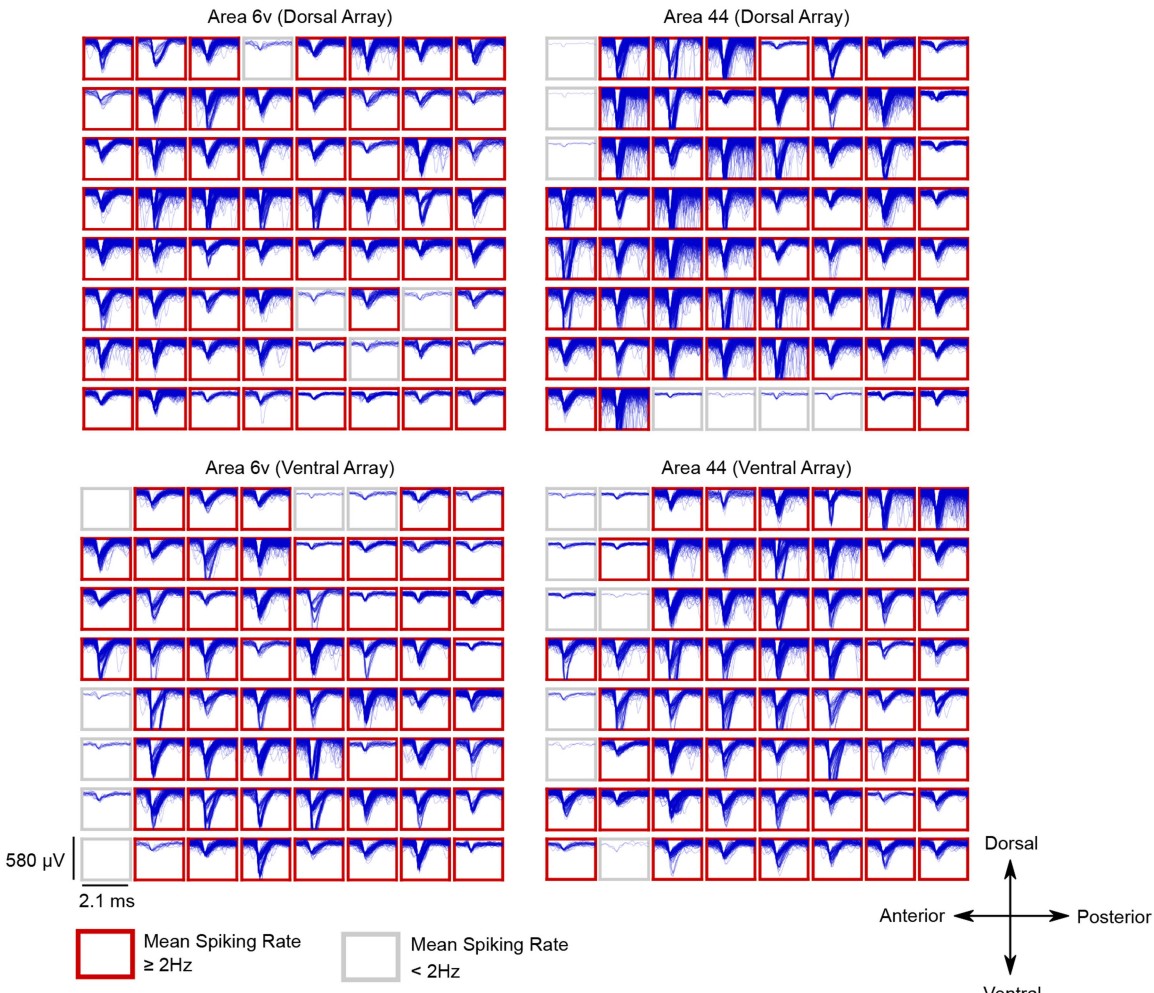

Area 6v (Dorsal Array)

Area 44 (Dorsal Array)

Area 6v (Ventral Array)

Area 44 (Ventral Array)

580 μV

2.1 ms

Mean Spiking Rate ≥ 2Hz

Mean Spiking Rate < 2Hz

Dorsal

Anterior ← → Posterior

Ventral

**Extended Data Fig. 1 | Example spiking activity recorded from each microelectrode array.** Example spike waveforms detected during a 10-s time window are plotted for each electrode (data were recorded on post-implant day 119). Each 8x8 grid corresponds to a single 64-electrode array, and each rectangular panel in the grid corresponds to a single electrode. Blue traces show example spike waveforms (2.1-ms duration). Neural activity was band-pass filtered (250–5000 Hz) with an acausal, 4th order Butterworth filter. Spiking events were detected using a −4.5 root mean square (RMS) threshold, thereby excluding almost all background activity. Electrodes with a mean threshold crossing rate of at least 2 Hz were considered to have 'spiking activity' and are outlined in red (note that this is a conservative estimate that is meant to include only spiking activity that could be from single neurons, as opposed to multiunit 'hash'). The results show that many electrodes record large spiking waveforms that are well above the noise floor (the y axis of each panel spans 580 μV, whereas the background activity has an average RMS value of only 30.8 μV). In area 6v, 118 electrodes out of 128 had a threshold crossing rate of at least 2 Hz on this particular day (113 electrodes out of 128 in area 44).

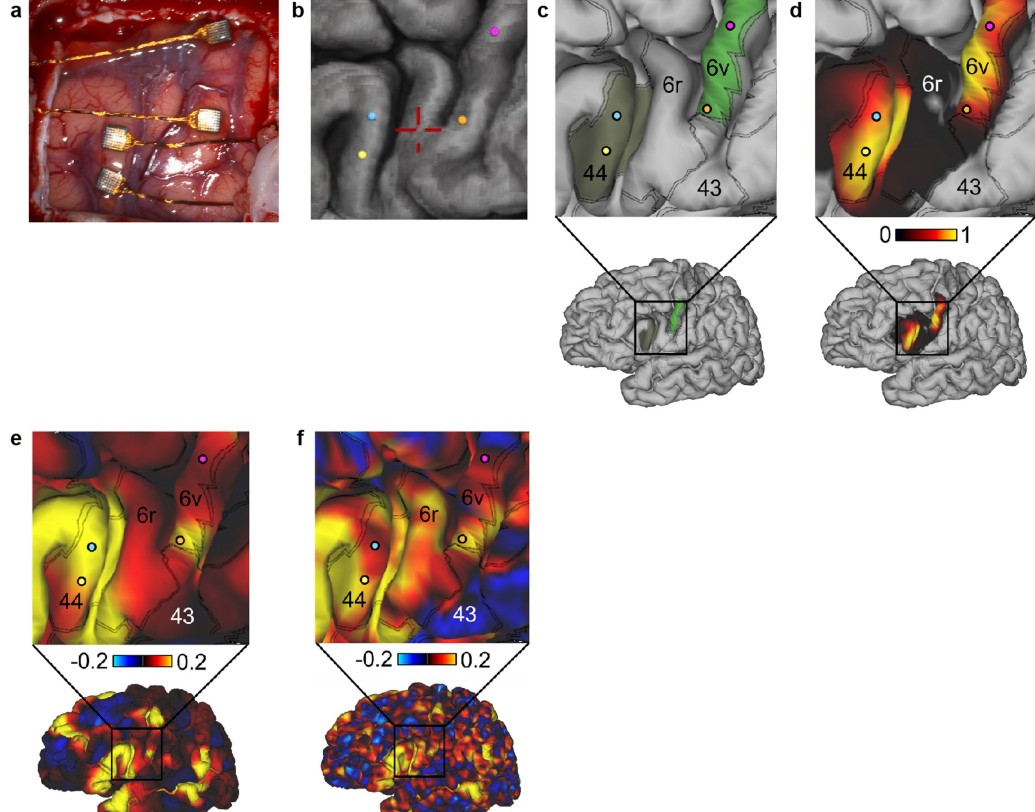

**Extended Data Fig. 2 | Array implant locations and fMRI data shown relative to HCP-identified brain areas.** (**a**) Array implants shown directly on the brain surface during surgery. (**b**) Array locations shown on a 3D reconstruction of the brain (array centers shown in blue, yellow, magenta, and orange circles) in StealthStation (Medtronic, Inc.). (**c**) approximate array locations on the participant's inflated brain using Connectome Workbench software, overlaid on the cortical areal boundaries (double black lines) estimated by the Human Connectome Project (HCP) cortical parcellation. (**d**) approximate array locations overlaid on the confidence maps of the areal regions. (**e**) A language-related resting state network identified in the Human Connectome Project data (N = 210) and aligned to T12's brain (**f**) the same resting state network shown in T12's individual scan. The ventral part of 6v appears to be involved in this resting state network, while the dorsal part is not. This resting state network includes language-related area 55b, Broca's area and Wernicke's area.

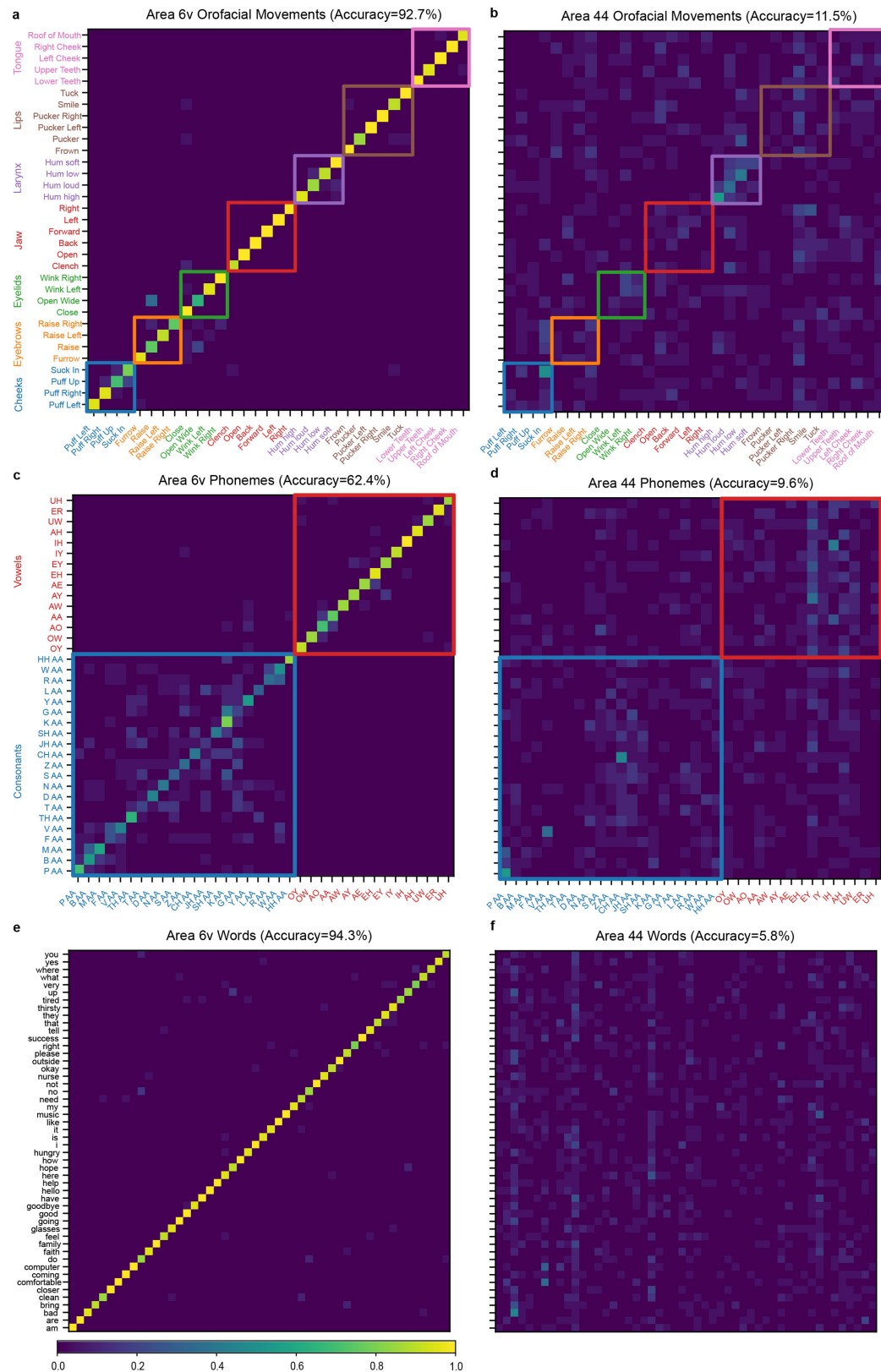

**Extended Data Fig. 3 | Classification confusion matrices for orofacial movements, individual phonemes, and individual words.** (**a**,**b**) Confusion matrices from a cross-validated, Gaussian naïve Bayes classifier trained to classify amongst orofacial movements using threshold crossing rates averaged in a window from 0 to 1000 ms after the go cue. Each entry (i, j) in the matrix is colored according to the percentage of trials where movement j was decoded (of all trials where movement i was cued). (**c**,**d**) Same as **a**,**b** but for individual phonemes. (**e**,**f**) Same as **a**,**b** but for individual words. Matrices on the left show results from using only electrodes in area 6v, while matrices on the right show results from using electrodes in area 44. Although good classification performance can be observed from area 6v, area 44 appears to contain little to no information about most movements, phonemes and words.

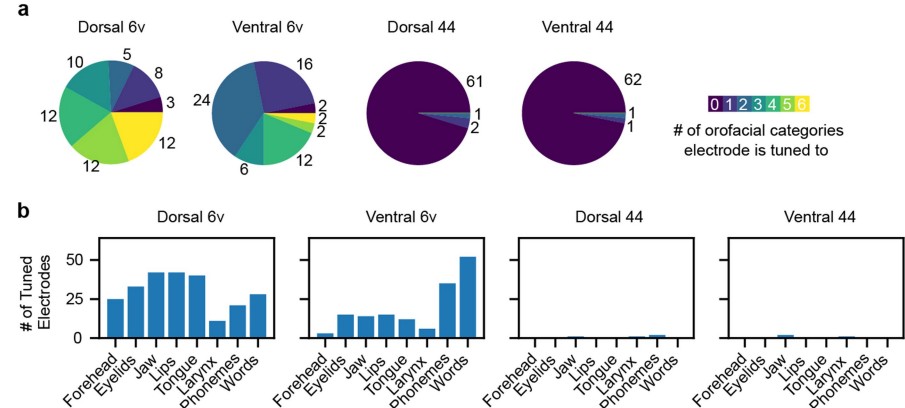

**Extended Data Fig. 4 | Individual microelectrodes are tuned to multiple categories of orofacial movement.** (a) Pie charts summarizing the number of electrodes that had statistically significant tuning to each possible number of movement categories (from 0 to 6), as assessed with a 1-way ANOVA (p < 1e-5). On the 6v arrays, many electrodes are tuned to more than one orofacial movement category (forehead, eyelids, jaw, lips, tongue, and larynx). (b) Bar plots summarizing the number of tuned electrodes to each movement category and each array. The ventral 6v array contains more electrodes tuned to phonemes and words, while the dorsal 6v array contains more electrodes tuned to orofacial movement categories. Nevertheless, both 6v arrays contain electrodes tuned to all categories of movement.

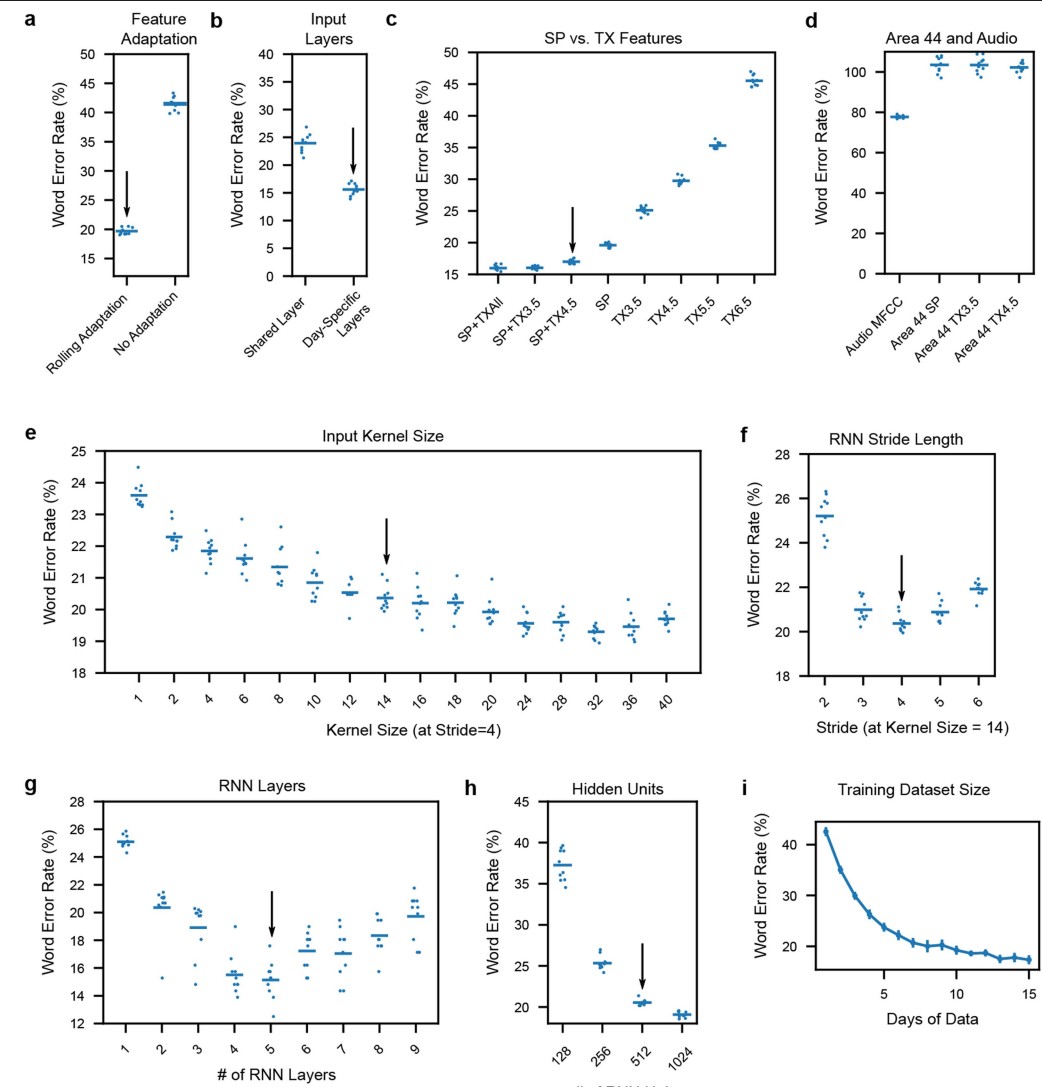

**Extended Data Fig. 5 | Offline parameter sweeps show the effect of RNN parameters and architecture choices.** Black arrows denote the parameters used for real-time evaluation. Blue open circles show the performance of single RNN seeds, while thin blue bars denote the mean across all seeds. (**a**) Rolling z-scoring improves performance substantially relative to no feature adaptation (when testing on held-out blocks that are separated in time from the training data). (**b**) Training RNNs with day-specific input layers improves performance relative to using a shared layer across all days. (**c**) RNN performance using different neural features as input (SP=spike band power, TX=threshold crossing). Combining spike band power with threshold crossings performs better than either alone. It appears that performance could have been improved slightly by using a −3.5 RMS threshold instead of −4.5. (**d**) RNN performance using audio-derived mel frequency cepstral coefficients ("audio

MFCC") or neural features from the area 44 arrays. While the MFCCs yield poor but above-chance performance, word error rates from IFG recordings appear to be at chance level (~100%). (**e**) RNN performance as a function of "kernel size" (i.e., the number of 20 ms bins stacked together as input and fed into the RNN at each time step). It appears that performance could have been improved by using larger kernel sizes. (**f**) RNN performance as a function of "stride" (a stride of *N* means the RNN steps forward only every *N* time bins). (**g**) RNN performance as a function of the number of stacked RNN layers. (**h**) RNN performance as a function of the number of RNN units per layer. (**i**) RNN performance as a function of the number of prior days included as training data. Performance improves by adding prior days, but with diminishing returns. The blue line shows the average word error rate across 10 RNN seeds and 5 evaluation days. Vertical lines show standard deviations across the 10 seeds.

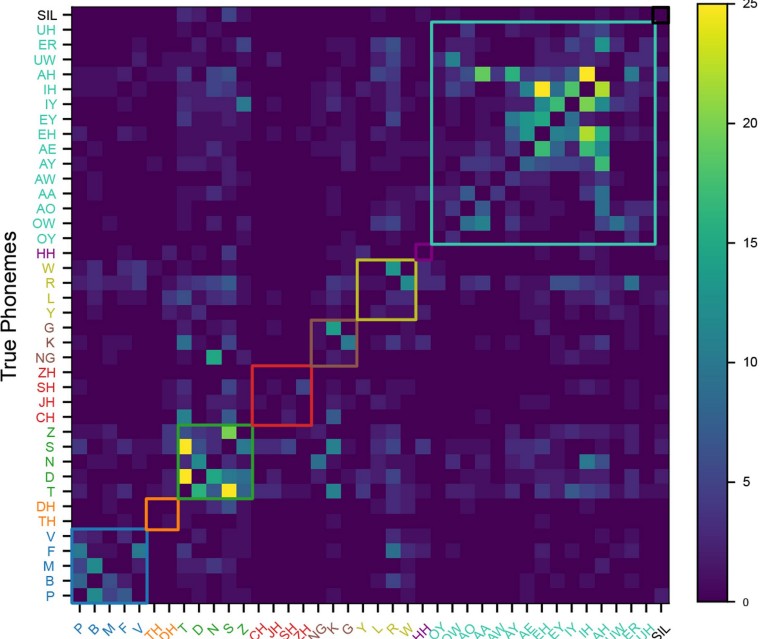

**Extended Data Fig. 6 | Phoneme substitution errors observed across all real-time evaluation sentences.** Entry *(i,j)* in the matrix represents the substitution count observed for true phoneme *i* and decoded phoneme *j*. Substitutions were identified using an edit distance algorithm that determines the minimum number of insertions, deletions, and substitutions required to make the decoded phoneme sequence match the true phoneme sequence. Most substitutions appear to occur between phonemes that are articulated similarly.

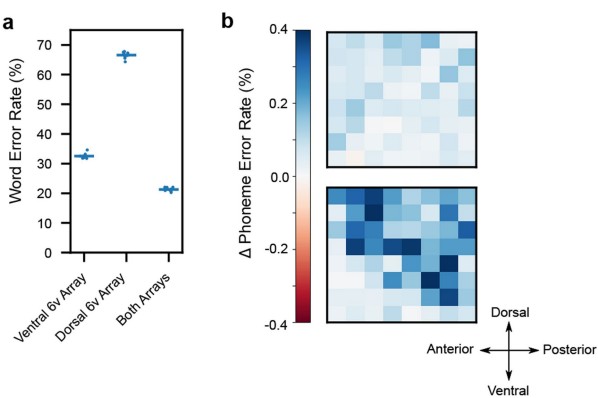

**Extended Data Fig. 7 | Contribution of each array and microelectrode to decoding performance.** (**a**) Word error rates for a retrospective offline decoding analysis using just the ventral 6v array (left column), dorsal 6v array (middle column), or both arrays (right column). Each circle indicates the word error rate for one of 10 RNN seeds. Word error rates were aggregated across 400 trials. Horizontal lines depict the mean across all 10 seeds. (**b**) Heatmaps depicting the (offline) increase in phoneme error rate when removing each electrode from the decoder by setting the values of its corresponding features to zero. Results were averaged across 10 RNN seeds that were originally trained to use every electrode. Almost all electrodes seem to contribute to decoding performance, although the most informative electrodes are concentrated on the ventral array. The effect of removing any one electrode is small (<1% increase in phoneme error rate), owing to the redundancy across electrodes.

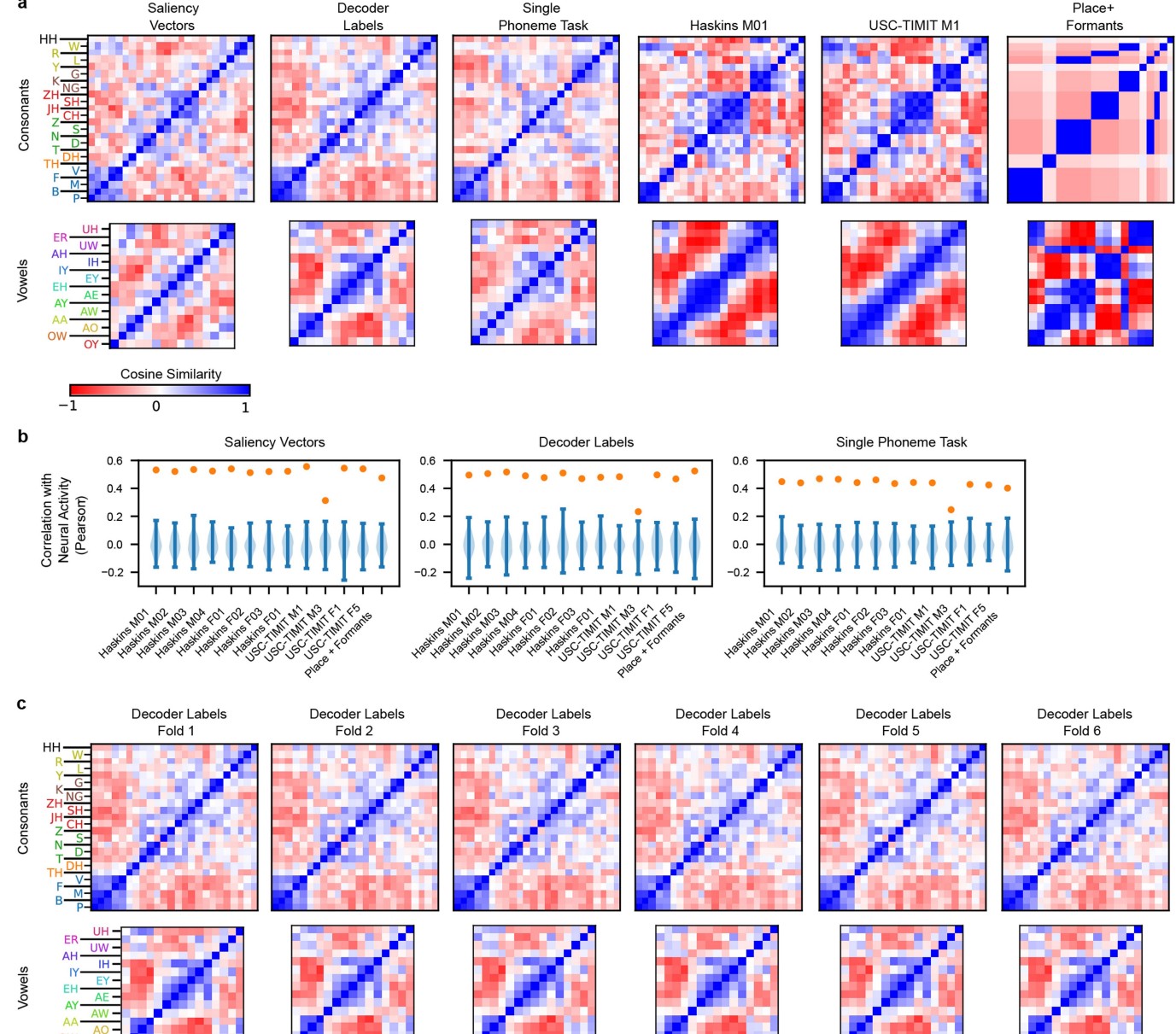

**Extended Data Fig. 8 | Additional methods and able-bodied subjects provide further evidence for an articulatory neural code. (a)** Representational similarity across consonants (top) and vowels (bottom) for different quantifications of the neural activity ("Saliency Vectors", "Decoder Labels", and "Single Phoneme Task") and articulator kinematics ("Haskins M01", "USC-Timit M1", "Place + Formants"). Each square in a matrix represents pairwise similarity for two phonemes (as measured by the cosine angle between the neural or articulatory vectors). Consonants are ordered by place of articulation (but with approximants grouped separately) and vowels are ordered by articulatory similarity (as measured by "USC-TIMIT M1"). These orderings reveal block-diagonal structure in the neural data that is also reflected in articulatory data. "Haskins M01" and "USC-Timit M1" refer to subjects M01 and M1 in the Haskins and USC-Timit datasets. "Place + Formants" refers to coding consonants by place of articulation and to representing vowels using their two formant frequencies. **(b)** Correlations between the neural representations and the articulator representations (each panel corresponds to one method of computing the neural representation, while each column corresponds to one EMA subject or the place/formants method). Orange dots show the correlation value (Pearson $r$), and blue distributions show the null distribution computed with a shuffle control (10,000 repetitions). In all cases, the true correlation lies outside the null distribution, indicating statistical significance. Correlation values were computed between consonants and vowels separately and then averaged together to produce a single value. **(c)** Representational similarity matrices computed using the "Decoder labels" method on 6 different independent folds of the neural data. Very similar representations across folds indicates that the representations are statistically robust (average correlation across folds = 0.79).

# Reporting Summary

## Statistics

For all statistical analyses, confirm that the following items are present in the figure legend, table legend, main text, or Methods section.

| n/a | Confirmed | |
|---|---|---|
| ☐ | ☒ | The exact sample size (*n*) for each experimental group/condition, given as a discrete number and unit of measurement |
| ☒ | ☐ | A statement on whether measurements were taken from distinct samples or whether the same sample was measured repeatedly |
| ☐ | ☒ | The statistical test(s) used AND whether they are one- or two-sided<br>*Only common tests should be described solely by name; describe more complex techniques in the Methods section.* |
| ☒ | ☐ | A description of all covariates tested |
| ☐ | ☒ | A description of any assumptions or corrections, such as tests of normality and adjustment for multiple comparisons |
| ☐ | ☒ | A full description of the statistical parameters including central tendency (e.g. means) or other basic estimates (e.g. regression coefficient) AND variation (e.g. standard deviation) or associated estimates of uncertainty (e.g. confidence intervals) |
| ☐ | ☒ | For null hypothesis testing, the test statistic (e.g. *F*, *t*, *r*) with confidence intervals, effect sizes, degrees of freedom and *P* value noted<br>*Give P values as exact values whenever suitable.* |
| ☒ | ☐ | For Bayesian analysis, information on the choice of priors and Markov chain Monte Carlo settings |
| ☒ | ☐ | For hierarchical and complex designs, identification of the appropriate level for tests and full reporting of outcomes |
| ☐ | ☒ | Estimates of effect sizes (e.g. Cohen's *d*, Pearson's *r*), indicating how they were calculated |

*Our web collection on statistics for biologists contains articles on many of the points above.*

## Software and code

Policy information about availability of computer code

| | |
|---|---|
| Data collection | The software for running th experimental tasks, recording data and real-time sentence decoding was a custom developed system using MATLAB, Simulink Real-Time, and Python. Software packages used included tensorflow 2.10.0, gp2_en 2.1.0, WeNet, SRILM and Kaldi. |
| Data analysis | Data was analyzed using custom MATLAB and Python code. Code is publicly available on GitHub here: https://github.com/fwillett/speechBCI |

For manuscripts utilizing custom algorithms or software that are central to the research but not yet described in published literature, software must be made available to editors and reviewers. We strongly encourage code deposition in a community repository (e.g. GitHub). See the Nature Portfolio guidelines for submitting code & software for further information.

## Data

Policy information about availability of data

All manuscripts must include a data availability statement. This statement should provide the following information, where applicable:
- Accession codes, unique identifiers, or web links for publicly available datasets
- A description of any restrictions on data availability
- For clinical datasets or third party data, please ensure that the statement adheres to our policy

All neural data needed to reproduce the findings in this study are publicly available on Dryad here: (link & DOI to be added - under review at Data Dryad now). The dataset contains neural activity recorded during the attempted speaking of 10,850 sentences, as well as instructed delay experiments designed to investigate the neural representation of orofacial movement and speech production. As part of this study, we also analyzed publicly available electromagnetic articulography data:

# Research involving human participants, their data, or biological material

Policy information about studies with <ins>human participants or human data</ins>. See also policy information about <ins>sex, gender (identity/presentation), and sexual orientation</ins> and <ins>race, ethnicity and racism</ins>.

| | |
|---|---|
| Reporting on sex and gender | This study included data from one participant, T12, who is a biological female and identifies as a woman. This information was self-reported. No sex or gender based analyses were performed given there was only a single participant and the study was assessing brain-computer interface performance. |
| Reporting on race, ethnicity, or other socially relevant groupings | This study assessed brain-computer interface performance for a single participant. No variables relating to race, ethnicity or other socially relevant groupings were reported or analyzed. |
| Population characteristics | This study includes data from one participant (identified as T12) who gave informed consent and was enrolled in the BrainGate2 Neural Interface System clinical trial (CliniclaTrials.gov Identifier: NCT00912041, registered June 3, 2009) but this study did not report clinical trial results. T12 is a left-handed woman, 67 years old during data collection with bulbar ALS that began approximately 9 years prior to enrollment. |
| Recruitment | Participant T12 was enrolled in the BrainGate 2 clinical trial after meeting inclusion criteria based in part on disease characteristics. Inclusion and exclusion criteria are available online (ClinicalTrials.gov). |
| Ethics oversight | The BrainGate2 Neural Interface System clinical trial was approved under an Investigational Device Exemption (IDE) by the US Food and Drug Administration (IDE #G09003). Permission was also granted by teh Institutional Review Board of Stanford University (protocol #20804). All research was performed in accordance with relevant guidelines/regulations. |

Note that full information on the approval of the study protocol must also be provided in the manuscript.

# Field-specific reporting

Please select the one below that is the best fit for your research. If you are not sure, read the appropriate sections before making your selection.

☒ Life sciences ☐ Behavioural & social sciences ☐ Ecological, evolutionary & environmental sciences

For a reference copy of the document with all sections, see nature.com/documents/nr-reporting-summary-flat.pdf

# Life sciences study design

All studies must disclose on these points even when the disclosure is negative.

| | |
|---|---|
| Sample size | No sample-size calculation was performed. Data were collected in a single participant to characterize the performance of a brain-computer interface. Uncertainty in performance estimates were quantified with confidence intervals, and show a robust result. |
| Data exclusions | This study is based on brain-computer interface performance evaluation data collected over a series of days. All days are reported in the study and all relevant data is included. |
| Replication | This study assessed brain-computer interface performance with a single participant. Results were replicated across eight independent days of performance evaluation. |
| Randomization | Randomization into groups is not relevant for this study as only one participant is included in the study. |
| Blinding | Blinding is not relevant to this study as only one participant was included to asses the performance of a brain-computer interface. |

# Reporting for specific materials, systems and methods

We require information from authors about some types of materials, experimental systems and methods used in many studies. Here, indicate whether each material, system or method listed is relevant to your study. If you are not sure if a list item applies to your research, read the appropriate section before selecting a response.

## Materials & experimental systems

| n/a | Involved in the study |
|-----|----------------------|
| ☒ | Antibodies |
| ☒ | Eukaryotic cell lines |
| ☒ | Palaeontology and archaeology |
| ☒ | Animals and other organisms |
| ☒ | Clinical data |
| ☒ | Dual use research of concern |
| ☒ | Plants |

## Methods

| n/a | Involved in the study |
|-----|----------------------|
| ☒ | ChIP-seq |
| ☒ | Flow cytometry |
| ☒ | MRI-based neuroimaging |

