## [Peer Review File · Nature]

Manuscript Title: A high-performance speech neuroprosthesis

Reviewer Comments & Author Rebuttals

Reviewer Reports on the Initial Version:

Referees' comments:

Referee #1 (Remarks to the Author):

The potential to restore speech communication to individuals who maintain language fluency but have lost motor speech (due to motor neuron disease, stroke, etc.) would be an important advance in an individually devastating problem area. In this manuscript, the authors present data on the restoration of approximately 60 word-per-minute, 90.9% (50-word) or 76.2% (125K word) accurate BCI decoded speech. The neural sensor is a pair of intracortical arrays in area 6v (pre-motor cortex) in a single human subject. The authors utilize the presence of spatial intermixing, together with RNN decoding, to achieve this result. The authors were able to decode 34 orofacial movements with 92.7 percent accuracy and 39 phonemes with 60% accuracy. Notably, two electrodes in area 44 (Broca's area) had little representation of speech-associated movement. A language model produced only modest performance improvement. The authors further provide a substantial supplementary results appendix, expanding on their findings, as described below. Overall, this is an important and significant paper. It represents a major advance over a previous study (Moses et al.), which utilized high density surface ECoG signals. Furthermore, beyond the technical advance, the data presented have important implications for the localization of motor speech activity, the value of intracortical vs ECoG BCI, and algorithms for motor speech decoding from cortical activity. I believe this is an important work that will be of interest to the broad scientific and neuroscience communities.

Comments:

1. The study is single-subject, which raises the issue of repeatability. The authors possess significant data, which could potentially be used to mitigate this shortcoming. It would be useful to evaluate and to present additional analysis from each of the four arrays (two in 6v, two in 44).

a. Did the two arrays in area 6v contribute equally? This is an important question to demonstrate that arrays implanted in 6v are likely to allow useful BCI speech restoration. Otherwise, one might conclude that the authors simply "got lucky" with the placement of one of the arrays, which may not occur in a second subject. For example, a confusion matrix like SF3 could be presented for each of the four arrays. Alternatively, the authors could present the relationship between electrode/unit number and offline accuracy, as the most informative electrodes/units are sequentially dropped.

b. The authors used rsfMRI to target the arrays. It would be useful to provide (in supplementary information) a figure which shows the final location of the arrays relative to these connectivity results. Again, this will help to determine the degree to which one can predict optimal array placement, which has been an ongoing challenge in the field.

c. The authors indicate that the arrays in area 44 did not record useful motor information (SF2-3). Were signals from these arrays recorded in other experiments, and did those arrays modulate in a significant way during task performance (though perhaps not motor speech production)? From a scientific standpoint, it would increase the impact of the study if an analysis were performed to compare the information contained in Area 6v and Area 44 arrays. Could it be that “Broca’s region” was missed with array placement (and how would one know), or are the authors justified in asserting that Broca’s area is not involved in motor speech/phoneme production?

2. On Page 2, line 51, what is meant by intermixing on the single electrode level? Is it anything more than each speech articulator lights up multiple electrode shanks on the array, though not uniquely, but that each phoneme creates a pattern that is consistent over time?

3. How was one second of neural population data selected? What is the time that the phoneme/orofacial movement was being produced? Can phonemes and orofacial movements be distinguished separately? For example, do phonemes appear as constructions of orofacial movements in a logical way (e.g. lip movements with labial consonants)? Were decodes for speech articulators and phonemes consistent?

4. My understanding is that English has 44 phonemes. Which were eliminated in the 39 tested (digraphs?) from the CMU dictionary? Were these eliminated phonemes not present in the words produced? Did the 39 phonemes fully represent all sounds in the test vocabularies?

5. Performance was evaluated with either vocalization or “mouthing” without vocalization. An important question is whether neuronal connections to muscles and sensory/proprioceptive feedback are required for encoding in 6V. Are decodes effective if imagined vocalization (no movement) is attempted, i.e., does there have to be movement for signals in 6V to encode phonemes/orofacial movements?

6. The authors make frequent comparison to Moses et al, in terms of classification accuracy. It appears that the region of cortex for the 6v electrodes is similar to that for “classification” in Moses et al. but different from the “speech-detection” areas used to detect participants attempts at speech. In this study, the method to detect onset of speech effort was somewhat unclear. Was the RNN emitting the phoneme probability every 80 ms able to provide timing information or was the computer monitor cue used not only during training but also during testing? T12 pressed a button to finalize decoding, but was there a signal to cue the attempt at speech?

7. In the calculation of word speed, how was the time of decode included?

8. There are two major differences between intracortical and ECoG data: (1) the density of electrodes, and (2) the information interpreted by the electrodes. If instead of threshold crossings, high gamma signaling is used as the basis of interpretation, how does the accuracy change? To fairly compare the two techniques, an offline comparison of high-gamma power could be compared to spiking data. Also, it appears that spiking band power was calculated but not used for interpretation. How did SBP compare with threshold crossing as the source of neural information?

Thank you for the opportunity to evaluate this fine manuscript. I hope the authors find these comments and suggestions useful in clarifying the findings and increasing the impact of their work.

Referee #2 (Remarks to the Author):

The team presents the first large vocabulary speech neuroprosthesis. For this purpose, they implanted a patient suffering from ALS with 2 microelectrode arrays each in 6v and area 44. Due to the disease, which was already diagnosed 8 years ago, the patient is indeed not able to produce intelligible speech. This is nicely demonstrated in two high-quality videos. With a total of 26 data recording sessions in different tasks, the researchers are able to show that neurons in 6v are nicely tuned to orofacial movements. The team then employs an RNN and a 3-gram language model in a Viterbi search to decode attempted speech (both audible and silent) with small and very large vocabularies.

This study presents a huge leap in the field of speech neuroprosthesis and will be of large interest to the wider scientific readership. It combines amazing data with sophisticated analysis and thereby produces results far above the current state of the art. To me, the most exciting highlights are:

- * First large vocabulary speech neuroprosthesis in a patient that is indeed anarthric. The communication speed and naturalness thereby exceed anything presented previously.
- * First implantation of Utah Arrays with the specific goal of creating a speech neuroprosthesis. This by itself is a huge feat. Interestingly, area 44 which is implicated in all theories of speech production did not provide a lot of useful information in this setup.
- * The neural decoding model from audible speech transferred almost perfectly to silently mouthed speech, highlighting how similar the underlying neural processes are despite the absence of laryngeal movements.

In addition to these highlights, the authors analyze several design aspects of speech neuroprostheses that can be of relevance for future (clinical) studies.

Overall, this article is very well written and presents a wonderful advance to the field. The authors write clearly and concisely without overclaiming and compare their results to the most relevant state of the art. Their conclusions are robustly underpinned by data and careful analysis. Naturally, they also appropriately credit the relevant previous work. I particularly enjoyed the description of the methods, which is easy to follow and does not try to obscure the procedure. Of course, it would be even better to share data and code, something I strongly suggest before publication of this work.

Some comments and suggestions remain before this article can be accepted however:

Larger points:

- (1) I am not convinced by the extrapolation of decoding results depending on the amount of electrodes in Fig. 4(b). The data in the graph sampling results depending on the number of used electrodes is of interest and important for future speech neuroprostheses. It is also clear to see that results have not saturated, yet. However, the extrapolation to even more electrodes makes a large

number of assumptions which might not be true. Can we really implant 10 times as many electrodes and still get more useful information? If we have electrodes of the same size, we might end up in areas that do not provide meaningful information anymore. If we use smaller electrodes, they might all measure the same signal. So I believe it would be good to show the data without extrapolation.

(2) For the same plot, it would be good to include standard deviation or standard errors across the 100 seeds for each amount of channels to indicate the variability. Same applies to Figure S4(i).

(3) The analysis for the preserved articulatory representation is not convincing to me (even though I fully believe that this is the case). The authors state that they do not have articulatory labels, as “we do not have ground truth knowledge of when each phoneme is being spoken” (line 146) and thus use saliency maps instead. They then choose some arbitrarily chosen EMA data of a single subject to compare against. I think this is not a very convincing analysis, as it purely analyzes what the neural network has learned (to best discriminate), as opposed to the actual neural data and compares it to some very arbitrary data. It would be much more straight-forward to simply use the trained RNN to provide phonemic labels on the data (which the authors describe how to do). Then one can easily use the IPA table to get the articulator movements during each point of time. This way, the authors would have a large sample of neural data for each articulator during attempted speech production and could do convincing statistics that the articulators are indeed still represented. This would be a lot more convincing than a cosine similarity of one vector per phoneme.

(4) I am a bit surprised that the Phoneme Error Rates in Table 1 are the same for the online results and the improved LM. The better language model should also improve the alignment of the phonemes and if the phoneme errors are exactly the same, how can it construct better words from it (as the WER is lower)? It would be great if the authors could check again.

Smaller:

(1) On p.3 line 75, the authors write “These probabilities were then processed by a language model to ...”. This is not described accurately: The probabilities are combined with the language model in the Viterbi search, the language model itself just supplies probabilities for sequences of words and does not process probabilities of the RNN.

Suggestions:

(1) I believe that for future BCI studies and eventually even clinical trials, it is far more important to know how much training data needs to be collected than how much data could be used for adaptation from each day. This is particularly true as no adaptation is the more realistic use case (as the authors write in line 470). So for me, it would be better to have S4(i) in the main manuscript instead of 4(c).

(2) For the section headings within the result section, I would prefer it if the main finding could be summarized instead of the analysis step. So instead of “Representation of orofacial movement”, I would prefer “Orofacial movements are represented in 6v but not in area 44”. Instead of “Decoding Attempted Speech”, I would prefer “Attempted Speech can be decoded even when silently mouthed”. Again, this is just a suggestion.

Referee #3 (Remarks to the Author):

I briefly respond to the items listed on the review page of nature, and list comments below

A Summary of the key results

results are rather overstated regarding relevance to BCI end users

B Originality and significance: if not novel, please include reference
work is highly original

C Data & methodology: validity of approach, quality of data, quality of presentation
more than adequate

D Appropriate use of statistics and treatment of uncertainties
adequate

E Conclusions: robustness, validity, reliability

more attention is required to address the limitations regarding generalizability, recalibrations and
longevity of functionality

F Suggested improvements: experiments, data for possible revision
see comments below

G References: appropriate credit to previous work?
adequate

H Clarity and context: lucidity of abstract/summary, appropriateness of abstract, introduction and
conclusions

Generally good but amendments required to tone down some overstated claims

My Review:

Authors report on a single participant with 4 Utah arrays on pre central and inferior frontal left
cortex. Words were decoded by assessing threshold crossing in 64 electrodes of one array in inferior
pre central cortex, yielding 62 words/minute at an error rate of 9-23% depending on used
vocabulary size. Data are reported for 8 days of testing

The patient could still produce some speech sounds albeit unintelligible, thus could still contract
articular muscles.

Overall, the work is novel and indeed for the first time displays proof of concept of decoding speech
for people with speech disabilities with Utah arrays. However, several issues dampen my
enthusiasm. The paper would be considerably stronger if authors would take the comments into
consideration and amend interpretation and most importantly conclusions

1) For one, multiple statements give readers the suggestion that the findings are of direct relevance
for clinical use which any opinion is overstated given the limited evidence. Most importantly, since
the participant can still move facial muscles involved in articulation (the patient makes obvious head
movements following the rhythm of the words in a sentence) which contribute to sensory feedback
and thus possibly to decoding (in the sensorimotor face area motor and sensory function is mixed in
pre-and post-central gyrus), it is by no means clear whether the findings apply to people with full
facial paralysis. In particular, the statement that 'Our demonstration is a proof of concept that
decoding attempted speaking movements from intracortical recordings is a promising approach' is

overly bold given that the participant was still capable of actual articulator/face movements. Even during silent speech actual movements were made ('mouthing the words') which means signals included sensory feedback plus actual motor output.

2) Second, recalibration is required every day (involving several hundred sentences) which, unless convincingly solved, is a significant drawback for daily use.

3) Third, longevity of Utah arrays function is a recurring source of concern for practical use in vulnerable end-users. Since Utah arrays have been reported to degrade within a few years, and since replacement requires resurgery and placement of new arrays in different locations due to scarring of the brain tissue, feasibility for longterm BCI use is not indicated sufficiently. Placing more arrays as suggested in the discussion will only aggravate the problem of short longevity and renders less brain tissue to place new arrays. These concerns in my opinion should be regarded carefully, and claims should be toned down accordingly.

4) Fourth, throughout the manuscript the performance is compared to previous papers repeatedly, notably a single paper by the Chang group. Given the speed at which performance is improved in general in speech decoding, such comparisons are not very informative and rather fleeting in nature. I strongly suggest to reduce such comparisons, since the paper is the first to show speech decoding with Utah arrays which should be sufficiently novel by itself without repeated comparisons to Chang's paper.

Other Comments:

5) More detailed exploration of the neural sources contributing to high decoding performance would strengthen the paper. Is it correct to conclude that different phoneme and word representations are distributed across the face area? How do the results shown in Figure 1c-d can be interpreted taking into account the somatotopic representation of speech articulators on the sensorimotor cortex (see Bouchard et al. 2013)?

6) After training did the participant need to calibrate the system by copying sentences every day?

7) Training of the RNN was done 'at the beginning of each day'. Does this refer to the 8 days of testing the decoders? Or does this refer to separate days prior to testing?

8) Authors report use of silence tokens, how were they determined? And was the participant instructed to pause between words?

9) Why did the participant need to press a button to indicate end of sentence production?

10) In Fig 4 b: extrapolating the line is very speculative given it is not clear whether actual additional electrodes will yield additional discriminatory information. This claim should be presented with less confidence also in the discussion

11) An improved offline decoding approach was reported that performed better when training and

test sentence were closer together in time. Authors report that better models that adapt to signal non-stationarity should be developed. Yet how would that work in real time without requiring the participant to frequently copy text to update training (which would render the BCI system cumbersome at least)

12) The authors state several times that they attribute the improvement in decoding performance to a much larger number of channels (page 6, lines 205-206, page 8 lines 248). It seems to me that the performance gains could rather be attributed to higher density or recordings and acquiring data at the single-neuron level. Can the authors clarify their position and address this more explicitly in the Discussion?

Figures

13) Support fig 1: legend does not explain what each figure represents. It is mentioned that each 8x8 matrix is one electrode but it seems to represent 64 electrodes on a single array. Moreover, mention is made of 128 electrodes without indicating they mean area 6v

14) To improve readability, I suggest that the authors add titles to individual plots on Figure 4S and axis labels to Figure 5S.

Referee #4 (Remarks to the Author):

Review of Willett et al. "A high-performance speech neuroprosthesis"

Summary of Key Results

This study reports on the performance of a intracortical microelectrode based speech-to-text BCI in a single participant with ALS. In the setting of a clinical trial protocol with IDE, the team implanted four 64-channel microelectrode arrays: two on "Broca's area" and two on orofacial motor cortex. Using state of the art modeling approaches, the team reports a 9.1% WER rate with a 50 word vocab and 23.8% WER with a 125,500 word vocab with speech decoded at 62 words per minute. The novel use of small high density microelectrode arrays shows spatially intermixed tuning to speech articulators, which provides evidence that speech BCIs can probably perform with less cortical coverage than previously described. I agree with the authors that the results demonstrate early feasibility of microelectrode-based BCIs and will pave the way for future clinical studies in larger groups of patients. The results are exciting.

Originality and Significance

This is a highly original first-of-a-kind study that uses microelectrode recordings from pars triangularis and orofacial motor cortex as a speech BCI. The major significance of the work is that it demonstrates early feasibility of this approach for BCI design. As the authors point out, the microelectrode recordings provide new information about the representation of orofacial movements at the single neuron level.

Data and Methodology

Standard implantation approach based on fMRI data used to lateralize language given that patient is historically L handed. 4 microelectrode arrays placed. Each training session was numbered in the methods.

Can authors specify how many hours were spent on each training session? This would be interesting, if not helpful, to really understand how much training these BCIs require of patients.

Appropriate Use of Statistics and Treatment of Uncertainties

Analytic approaches seem appropriate to me.

Conclusions

The authors conclude that this work suggests that a higher channel count system that records from a small area 6v is a feasible path forward that will restore conversational speeds to people with paralysis. This is a reasonable conclusion of the work, though as the authors point out earlier, there is much more work to be done. I think it is appropriate that the authors temper their conclusions about how this approach compare's with Chang's ECoG-based approach (which they cite from NEJM) given that both approaches could yield clinically-valuable results in future studies.

Suggested Improvements

Pg 2. Lines 45-46: I think you mean to say "ventral motor cortex" not ventral premotor cortex.

Pg 2. Second paragraph: it is unclear here what task is being performed to generate phonemes. Is this the same as the orofacial movement task described in the prior paragraph? I think so based on reading the methods, but some clarity would be helpful in the main text.

Pg3. 118 – 119 – I agree with the advantages of intracortical MER, however, since there are so many other factors that influence decodability, I'm not sure the authors should make blanket statements about this being responsible for their increased decodability result.

I would make it clear in the main text exactly how much training (in hours) to perform each of the decoding tasks.

References

References seems restricted to a small number of groups consider citing more work by Kai Miller, Nathan Crone, Eddie Chang...etc.

Author Rebuttals to Initial Comments:

Reply to Reviewers

Note: reviewers' comments appear in **black text**. Our replies appear in **blue text**, and revised manuscript text appears indented (with old text shown in **black** and new edits in **red**).

Overview:

We thank the reviewers for their careful read of the manuscript and their insightful and helpful suggestions. We are gratified to see the reviewers' unanimous recognition that this work represents a substantial advance in brain-computer interfaces and the neuroscience of speech. In much-appreciated support of our work, summary comments included: Reviewer 1 (R1), "Overall, this is an important and significant paper. I believe this is an important work that will be of interest to the broad scientific and neuroscience communities." R2, "This study presents a huge leap in the field of speech neuroprosthesis and will be of large interest to the wider scientific readership. It combines amazing data with sophisticated analysis and thereby produces results far above the current state of the art." R3, "Overall, the work is novel and indeed for the first time displays proof of concept of decoding speech for people with speech disabilities with Utah arrays." R4, "the results demonstrate early feasibility of microelectrode-based BCIs and will pave the way for future clinical studies in larger groups of patients. The results are exciting."

Most of the questions raised were requests for further clarification, reframing/interpreting results with more caution, and suggestions for additional analyses. We have addressed all of these requests, which we believe has improved the presentation, rigor and impact of the work. Below, we give a brief overview of the four questions that resulted in new figures and/or significant changes to the presentation of the work. Point-by-point responses to each reviewer suggestion appear further below this higher-level "Overview" section.

First, R2 asked for additional evidence demonstrating a preserved articulatory representation of phonemes in area 6v. We implemented new ways of showing that the neural activity contains a preserved representation (including what R2 suggested). We include these in a new supplemental figure showing that the results of our analysis are robust to different ways of quantifying the neural and articulatory representations.

Second, R1 asked us to determine whether one of the two arrays placed in area 6v contained more information about speech than the other. In doing these additional analyses, we uncovered an indexing error in the code for our original somatotopy analyses. The corrected somatotopy maps (in addition to several new analyses) now reveal interesting differences between the two 6v arrays, where the ventral array does indeed appear to be more speech-specific and the dorsal array appears more tuned to orofacial movement. Nevertheless, our original main result still holds: both arrays are tuned to speech and orofacial movement, and both arrays have highly intermixed somatotopy. Our new results are consistent with resting state fMRI data from the Human Connectome Project [1] and from T12 that situates the ventral region of 6v as part of a language-related network, and we think this should strengthen the neuroscience impact of this work. However, future work must be done to see if these differences between the dorsal and ventral part of area 6v are consistent across participants.

Third, R3 and R4 asked us to minimize comparisons to the most recent ECoG speech decoding work [2]. We have reduced the number of comparisons to the Chang lab work, and re-worded the conclusion as to how the intracortical approach differs from ECoG. Nevertheless, we think it is important to compare to the start of the art, and to let the reader know how these performance numbers fit within the broader context of the field. Therefore, we left two comparisons in the manuscript: one in the abstract, and one in the results section when describing the performance numbers.

Fourth, R3 asked several questions about the clinical viability of Utah microelectrode arrays. Several longstanding obstacles remain for clinically translating intracortical systems, including: (1) longevity of the recording technology, (2) adaptation to signal nonstationarity without burdening the user with frequent retraining, and (3) a need for testing the efficacy of these systems in fully locked-in people. As such, we have done our best to make it clear that we see the current work only as a “proof of principle” that high-performance neural decoding of attempted speech is possible using intracortical arrays. We do not see the work as a complete, clinically viable system in its current form, and have re-worded the discussion to make this clearer. Nevertheless, we believe that our study provides important motivation for the field to continue working on these problems, since it demonstrates several key principles which will be beneficial for a future clinical device: successful large-vocabulary decoding, intermixed tuning within a small cortical area, and a preserved representation of speech in a person with severe speech impairment due to paralysis. We are optimistic that these well-known challenges with intracortical technologies can and will be overcome, with studies such as this one serving as the additional motivation to do so.

[1] Glasser, M. F. *et al.* A multi-modal parcellation of human cerebral cortex. *Nature* **536**, 171 (2016).

[2] Moses, D. A. *et al.* Neuroprosthesis for Decoding Speech in a Paralyzed Person with Anarthria. *New England Journal of Medicine* **385**, 217–227 (2021).

Referee #1 (Remarks to the Author):

The potential to restore speech communication to individuals who maintain language fluency but have lost motor speech (due to motor neuron disease, stroke, etc.) would be an important advance in an individually devastating problem area. In this manuscript, the authors present data on the restoration of approximately 60 word-per-minute, 90.9% (50-word) or 76.2% (125K word) accurate BCI decoded speech. The neural sensor is a pair of intracortical arrays in area 6v (pre-motor cortex) in a single human subject. The authors utilize the presence of spatial intermixing, together with RNN decoding, to achieve this result. The authors were able to decode 34 orofacial movements with 92.7 percent accuracy and 39 phonemes with 60% accuracy. Notably, two electrodes in area 44 (Broca's area) had little representation of speech-associated movement. A language model produced only modest performance improvement. The authors further provide a substantial supplementary results appendix, expanding on their findings, as described below. Overall, this is an important and significant paper. It represents a major advance over a previous study (Moses et al.), which utilized high density surface ECoG signals. Furthermore, beyond the technical advance, the data presented have important implications for the localization of motor speech activity, the value of intracortical vs ECoG BCI, and algorithms for motor speech decoding from cortical activity. I believe this is an important work that will be of interest to the broad scientific and neuroscience communities.

We are gratified that the reviewer felt that this paper is a major advance that should attract broad interest. We thank the reviewer for their thorough read of the manuscript and insightful and helpful questions and suggestions.

Comments:

1. The study is single-subject, which raises the issue of repeatability. The authors possess significant data, which could potentially be used to mitigate this shortcoming. It would be useful to evaluate and to present additional analysis from each of the four arrays (two in 6v, two in 44).

Thank you for these helpful suggestions, which we did our best to address below. We added new analyses and figures which we believe strengthen the work. Additionally, after examining the somatotopy and array differences more closely, we identified and corrected an indexing error in our code which generated the original array somatotopy maps. The corrected maps now reveal differences between the two 6v arrays (the ventral array now appears to be more speech-specific). We are grateful to the reviewer for asking us to investigate this in more detail!

While the main result still holds (both arrays are tuned to both speech and orofacial movement, and both have highly intermixed somatotopy), the new results suggest that the upper and lower parts of 6v are not identical in function. This finding is consistent with resting state fMRI data from the Human Connectome Project [1] and from T12 that situates the ventral region of 6v as part of a language-related network. We now include this new resting state data in a supplemental figure (see our responses below).

Whether this result will hold in additional participants can only be answered in future work. We now call attention to this issue in the discussion with the following new sentence:

Variability in brain anatomy is also a potential concern, and more work must be done to confirm that regions of precentral gyrus containing speech information can be reliably targeted.

Nevertheless, we think it is reasonable to suppose that arrays targeted to the ventral 6v area will show similarly speech-rich information in other participants, especially since ventral 6v clearly has different functional connectivity properties than dorsal 6v (as demonstrated by group averages from the Human Connectome Project, N=210). The inclusion of ventral 6v in a language-related resting state network was also identifiable in T12's individual resting state fMRI scan.

[1] Glasser, M. F. *et al.* A multi-modal parcellation of human cerebral cortex. *Nature* **536**, 171 (2016).

a. Did the two arrays in area 6v contribute equally? This is an important question to demonstrate that arrays implanted in 6v are likely to allow useful BCI speech restoration. Otherwise, one might conclude that the authors simply “got lucky” with the placement of one of the arrays, which may not occur in a second subject. For example, a confusion matrix like SF3 could be presented for each of the four arrays. Alternatively, the authors could present the relationship between electrode/unit number and offline accuracy, as the most informative electrodes/units are sequentially dropped.

Thank you for these suggestions. We added new figures to address this point directly. First, in Figure 1, we now report classification results by array (and by time) across orofacial movements, phonemes and words (Fig. 1e). These new results show interesting differences between the two 6v arrays. The ventral array appears to contain more information about speech, especially during the instructed delay / cue presentation period, while the dorsal array appears to contain more information about orofacial movements.

(e) Classification accuracy across time for each of the four arrays and three types of movement. Classification was performed with a 100 ms window of neural population activity beginning at each time point. Shaded regions show 95% confidence intervals. Gray lines show the time of speech onset (average sound envelope recorded by a microphone). Vertical dashed lines show the beginning of the delay and go periods. Horizontal dashed lines show chance classification performance.

These results are echoed in the corrected somatotopy maps in Fig. 1f, reproduced below:

(f) Tuning heatmaps for both arrays in area 6v, for each movement category. Circles are drawn if binned firing rates on that electrode are significantly different across the given set of conditions ($p < 1e-5$ assessed with a one-way ANOVA; bin width=800 ms). Shading indicates the fraction of variance accounted for by the across-condition differences in mean firing rate.

Finally, we added a new supplemental figure to directly analyze which array contributed the most towards RNN decoding of speech, reproduced below:

Supplemental Figure 7. (a) Word error rates of a retrospective offline decoding analysis using just the ventral 6v array (left column), dorsal 6v array (middle column), or both arrays (right column). (b) Heatmaps depicting the (offline) increase in phoneme error rate when removing each electrode from the decoder by setting the values of its corresponding features to zero. Almost all electrodes seem to contribute to decoding performance, although the most informative electrodes are concentrated on the ventral array. The effect of removing any one electrode is small (<1% increase in phoneme error rate), owing to the redundancy across electrodes.

This figure shows that while the ventral array alone outperforms the dorsal array, both arrays combined perform substantially better (32% WER from the ventral array decreases to 21% WER). The electrode contribution heatmaps show that a portion of electrodes on the ventral array appear to contribute the most, although contributions come from both arrays and a wide range of electrodes.

In sum, we believe these new analyses add new scientific insight to the work and call for future investigation of these apparent topographic differences in function, but nevertheless do show that rich speech information is present in both upper and lower areas of 6v.

Although these results raise the possibility of many new neuroscience-focused analyses and experiments, we think that these figures strike a good balance between raising a salient point about array differences and keeping the cohesive focus of the paper on BCI decoding of speech.

b. The authors used rsfMRI to target the arrays. It would be useful to provide (in supplementary information) a figure which shows the final location of the arrays relative to these connectivity results. Again, this will help to determine the degree to which one can predict optimal array placement, which has been an ongoing challenge in the field.

Thank you for this suggestion. Note that we used the human connectome project (HCP) multi-modal parcellation procedure to parcellate our participant's cortex into different brain areas [1], which uses more than just resting state fMRI data alone. The HCP parcellation procedure combines resting state fMRI and anatomical data (folding patterns, myelination) with an atlas of brain area characteristics derived from 210 human subjects. To our knowledge, it is the most sophisticated method available for targeting particular brain areas using subject-specific scans and produces a probability heatmap for each brain area. We added a supplemental figure (reproduced below) to include an image of these heatmaps for area 6v and area 44, along with the array locations and surgical images, which gives a sense of how confident we can be about brain area targeting.

We also include a panel that shows the existence of a language-related resting state network (identified by the HCP parcellation procedure, see supplemental figure 8a of [1]) that includes the ventral region of 6v but not the dorsal region. The clear existence of this network in the HCP data (210 subjects) suggests that the speech-specificity of ventral 6v is a robust phenomenon that ought to occur in future participants as well.

Encouragingly, this language-related resting state network can even be seen in T12's individual scans. Regardless, it is worth noting that the visibility of this network on T12's scans is not necessarily required for array placement targeting, since the HCP parcellation procedure uses many other resting state networks combined with myelination and folding data to produce an estimate of ventral 6v's location.

Supplemental Figure 2. Locations of the implanted arrays within areas 6v and 44 are shown (A) directly on the participant's brain surface and (B) on a 3D reconstruction of the brain and the implanted arrays (array centers shown in blue, yellow, magenta, and orange circles) in StealthStation (Medtronic, Inc.). (C) shows the approximate array locations in the participant's inflated brain using Connectome Workbench software, overlaid on the cortical areal boundaries (double black lines) estimated by the Human Connectome Project (HCP) cortical parcellation. (D) shows the approximate array locations on the same brain as (C) overlaid on the confidence maps of the areal regions. (E,F) A language-related resting state network identified in the Human Connectome Project data aligned to T12's brain (E) and in T12's individual scan (F). The ventral part of 6v appears to be involved in this resting state network, while the dorsal part is not. This resting state network includes language-related area 55b, Broca's area and Wernicke's area.

[1] Glasser, M. F. *et al.* A multi-modal parcellation of human cerebral cortex. *Nature* **536**, 171 (2016).

c. The authors indicate that the arrays in area 44 did not record useful motor information (SF2-3). Were signals from these arrays recorded in other experiments, and did those arrays modulate in a significant way during task performance (though perhaps not motor speech production)? From a scientific standpoint, it would increase the impact of the study if an analysis were performed to compare the information contained in Area 6v and Area 44 arrays. Could it be that "Broca's region" was missed with array placement (and how would one know), or are the authors justified in asserting that Broca's area is not involved in motor speech/phoneme production?

Thank you for raising these questions. The question of what kind of information area 44 contains, and how exactly it compares to 6v, is interesting and far-reaching. We believe these important questions are best answered in a separate manuscript, which would have the space to dive deeply into these questions across an array of different tasks (e.g., cognitive decision-making tasks). In our view, focusing on how these areas represent speech production, and the BCI implications thereof, is where the current manuscript can have the most impact. The new panels that we added to Figure 1 we think now better highlight the differences between area 6v and 44 with respect to orofacial movement and speech production:

(d) Classification accuracy of a naïve Bayes decoder applied to 1 second of neural population activity from area 6v (red bars) or area 44 (purple bars) across all movement conditions (34 orofacial movements, 39 phonemes, 50 words). Black lines show 95% confidence intervals. **(e)** Classification accuracy across time for each of the four arrays and three types of movement. Classification was performed with a 100 ms window of neural population activity for each time point. Shaded regions show 95% confidence intervals.

We have used the most accurate targeting method we are aware of to locate area 44, but future work must be done in additional participants to confirm this result. It is worth noting that the absence of production-related neural activity in area 44 is consistent with some recent work questioning the traditional role of Broca's area in speech production^{16–19}.

16. Tate, M. C., Herbet, G., Moritz-Gasser, S., Tate, J. E. & Duffau, H. Probabilistic map of critical functional regions of the human cerebral cortex: Broca's area revisited. *Brain* **137**, 2773–2782 (2014).

17. Flinker, A. *et al.* Redefining the role of Broca's area in speech. *Proc. Natl. Acad. Sci.* **112**, 2871–2875 (2015).

18. Gajardo-Vidal, A. *et al.* Damage to Broca's area does not contribute to long-term speech production outcome after stroke. *Brain* **144**, 817–832 (2021).

19. Andrews, J. P. *et al.* Dissociation of Broca's area from Broca's aphasia in patients undergoing neurosurgical resections. *J. Neurosurg.* **138**, 847–857 (2022).

2. On Page 2, line 51, what is meant by intermixing on the single electrode level? Is it anything more than each speech articulator lights up multiple electrode shanks on the array, though not uniquely, but that each phoneme creates a pattern that is consistent over time?

Thank you for raising these questions. By intermixing, we mean that many electrodes are tuned to multiple categories of orofacial movement, and that there appears to be no somatotopic organization of orofacial movements within or across the arrays. We added a supplemental figure to clarify these points and make them more quantitative:

Supplemental Figure 4. (a) Pie charts summarizing the number of electrodes that had statistically significant tuning to each possible number of movement categories (from 0 to 6), as assessed with a 1-way ANOVA ($p < 0.0001$). On the 6v arrays, many electrodes are tuned to more than one orofacial movement category (forehead, eyelids, jaw, lips, tongue, larynx). (b) Bar plots summarizing the number of tuned electrodes to each movement category and each array. The ventral 6v array contains more electrodes tuned to phonemes and words, while the dorsal 6v array contains more electrodes tuned to orofacial movement categories. Nevertheless, both 6v arrays contain electrodes tuned to all categories of movement.

3. How was one second of neural population data selected? What is the time that the phoneme/orofacial movement was being produced? Can phonemes and orofacial movements be distinguished separately? For example, do phonemes appear as constructions of orofacial movements in a logical way (e.g. lip movements with labial consonants)? Were decodes for speech articulators and phonemes consistent?

Thank you for these helpful questions. We selected one second of neural population activity since this is the time when most information about the movement occurs. We designed the new Figure 1e panels to show this point. We also added speech volume (gray lines) to show the time at which the movements occurred:

(e) Classification accuracy across time for each of the four arrays and three types of movement. Classification was performed with a 100 ms window of neural population activity beginning at each time point. Shaded regions show 95% confidence intervals. Gray lines show the time of speech onset (average sound envelope recorded by a microphone). Vertical dashed lines show the beginning of the delay and go periods. Horizontal dashed lines show chance classification performance.

We appreciate that how phonemes are related to orofacial movements is an interesting scientific question, but potentially a deep and involved one that we would prefer to tackle in future publications. The dissociation in tuning between orofacial/speech tuning strength between the dorsal and ventral arrays in

6v does potentially suggest, however, that the representation of phonemes is not purely the sum of their constituent articulator movements.

4. My understanding is that English has 44 phonemes. Which were eliminated in the 39 tested (digraphs?) from the CMU dictionary? Were these eliminated phonemes not present in the words produced? Did the 39 phonemes fully represent all sounds in the test vocabularies?

No phonemes were removed from the CMU dictionary, which was designed to use 39 phonemes. To our knowledge, there are different ways of parcellating the continuous phenomenon of spoken English into discrete phonemes, and thus the number of phonemes can vary depending on the system used.

5. Performance was evaluated with either vocalization or “mouthing” without vocalization. An important question is whether neuronal connections to muscles and sensory/proprioceptive feedback are required for encoding in 6V. Are decodes effective if imagined vocalization (no movement) is attempted, i.e., does there have to be movement for signals in 6V to encode phonemes/orofacial movements?

Thank you for raising these important points. We appreciate the concern about whether this result will generalize to cases where no sensory/proprioceptive feedback is available. In our view, the only way to know for sure would be to replicate the work in additional participants who have lost all ability to move their speech articulators; as such, we now explicitly mention this limitation with the following additional sentence in the discussion:

Furthermore, the decoding results shown here must be confirmed in additional participants, and their generalizability to people with more profound orofacial weakness remains an open question.

Nevertheless, we believe that the case shown here (where some sensory feedback is intact) is equally important, if not more so. In our experience, paralysis is often incomplete in some way. This is supported by a recent study of potential BCI users [1] that found that “incomplete” locked-in syndrome, which still prevented normal communication due to paralysis, was significantly more common than complete locked-in syndrome. Of those surveyed in [1], the majority had some remaining mouth/head/face movement.

Second, prior work with Utah array recordings in arm/hand area in people with more complete arm/hand paralysis has demonstrated a strong neural representation for attempted movement in the absence of overt movement (e.g., [2-5]). We are optimistic that a similar result will hold for orofacial area of motor cortex.

Finally, the reviewer brings up the question of imagined movement. We believe that *imagined* movement is a different phenomenon from *attempted* movement with complete paralysis. In both cases, no overt movement occurs. The difference is that the *intent* to move is preserved in the attempted case, but not in the imagined case. Prior work has shown that imagined movement elicits weaker representations than attempted movement [6,7]. In our experience, BCI performance is best in the attempted movement case, and we instruct all of our participants to attempt to move instead of simply imagining movement. For this reason, we elected not to explore imagined movement in this work. While this is an interesting scientific question, its relevance for BCI systems is perhaps less clear.

[1] Pels, Elmar G.M., Erik J. Aarnoutse, Nick F. Ramsey, and Mariska J. Vansteensel. “Estimated Prevalence of the Target Population for Brain-Computer Interface Neurotechnology in the Netherlands.” *Neurorehabilitation and Neural Repair* 31, no. 7 (July 2017): 677–85. <https://doi.org/10.1177/1545968317714577>.

[2] Collinger, J. L. *et al.* High-performance neuroprosthetic control by an individual with tetraplegia. *The Lancet* 381, 557–564 (2013).

[3] Simeral, J. D., Kim, S.-P., Black, M. J., Donoghue, J. P. & Hochberg, L. R. Neural control of cursor trajectory and click by a human with tetraplegia 1000 days after implant of an intracortical microelectrode array. *Journal of Neural Engineering* 8, 025027 (2011).

[4] Willett, F. R. *et al.* Hand Knob Area of Premotor Cortex Represents the Whole Body in a Compositional Way. *Cell* (2020) doi:[10.1016/j.cell.2020.02.043](https://doi.org/10.1016/j.cell.2020.02.043).

[5] Hochberg, L. R. *et al.* Neuronal ensemble control of prosthetic devices by a human with tetraplegia. *Nature* **442**, 164–171 (2006).

[6] Vargas-Irwin, C. E. *et al.* Watch, Imagine, Attempt: Motor Cortex Single-Unit Activity Reveals Context-Dependent Movement Encoding in Humans With Tetraplegia. *Frontiers in Human Neuroscience* **12**, (2018).

[7] Rastogi, A. *et al.* Neural Representation of Observed, Imagined, and Attempted Grasping Force in Motor Cortex of Individuals with Chronic Tetraplegia. *Sci Rep* **10**, 1429 (2020).

6. The authors make frequent comparison to Moses et al, in terms of classification accuracy. It appears that the region of cortex for the 6v electrodes is similar to that for “classification” in Moses et al. but different from the “speech-detection” areas used to detect participants attempts at speech. In this study, the method to detect onset of speech effort was somewhat unclear. Was the RNN emitting the phoneme probability every 80 ms able to provide timing information or was the computer monitor cue used not only during training but also during testing? T12 pressed a button to finalize decoding, but was there a signal to cue the attempt at speech?

Thank you for raising this lack of clarity. Neural decoding was triggered to begin automatically in sync with the computer monitor cue. We added the following clarification to the Results section:

After training, the RNN was evaluated in real-time on held-out sentences that were never duplicated in the training set. For each sentence, T12 first prepared to speak the sentence during an instructed delay period. When the go cue was given, neural decoding was automatically triggered to begin. As T12 attempted to speak, neurally decoded words appeared on the screen in real-time reflecting the language model’s current best guess (SVideo 1). When T12 was finished speaking, she pressed a button to finalize the decoded output.

7. In the calculation of word speed, how was the time of decode included?

Thank you for pointing out this lack of detail. Words per minute was defined as the total number of words spoken divided by the total speaking time, which for each trial was the time between the go cue and when the participant pressed the button to indicate that she was finished speaking. There was no additional delay required to finalize the decoder output, which completed within a single 80 ms time step after the button was pressed. To make this information easier to find in the methods, we created a separate section title “Words per minute” as follows:

4.2 Words per minute

Words per minute was defined as the number of words spoken divided by the total amount of speaking time. Speaking time for each trial was defined as the time from which the cue turned green to when the participant pushed the button to signal she had completed saying the prompted sentence. There was no additional delay required to finalize the decoder output, which completed within a single 80 ms time step after the button was pressed.

Confidence intervals for words per minute were computed via bootstrap resampling over individual trials and then re-computing the speaking rate over the resampled distribution (10,000 resamples).

8. There are two major differences between intracortical and ECoG data: (1) the density of electrodes, and (2) the information interpreted by the electrodes. If instead of threshold crossings, high gamma signaling is used as the basis of interpretation, how does the accuracy change? To fairly compare the two techniques, an offline comparison of high-gamma power could be compared to spiking data. Also, it

appears that spiking band power was calculated but not used for interpretation. How did SBP compare with threshold crossing as the source of neural information?

Thank you for these interesting suggestions. We appreciate the desire to compare more directly to ECoG data. However, high gamma power recorded from microelectrodes may not carry the same signal as high gamma power recorded from much larger ECoG contacts (even if they are placed in the same area of cortex). If high gamma power were found to be very informative on microelectrodes, it may not translate to ECoG contacts. Likewise, if high gamma power were not to contain much information, this would not necessarily be expected to generalize to ECoG. Therefore, we think the best we can do is to focus here on the features that are best for microelectrodes (spike band power and threshold crossings) and compare performance to prior published work on ECoG. Future work with denser ECoG grids will have to be done for a more direct comparison of the two modalities.

Note that we included both threshold crossing features and spike band power features in the decoder. Supplemental figure 4c examines (offline) how the decoding performance changes as different features are used and combined:

Here, SP=spike band power and TX=threshold crossing. Combining spike band power with threshold crossings performs better than either one alone, and it appears that performance could have been improved slightly by using a -3.5 RMS threshold instead of -4.5.

In the methods section 1.4, we state:

For decoding, threshold crossing counts and spike band power from the 128 electrodes in area 6v were concatenated to yield a 256 x 1 feature vector per time step. For neural tuning analyses (e.g. Figure 1), only threshold-crossing counts were used.

Thank you for the opportunity to evaluate this fine manuscript. I hope the authors find these comments and suggestions useful in clarifying the findings and increasing the impact of their work.

Thank you again for your helpful comments, which we believe have significantly improved the paper!

Referee #2 (Remarks to the Author):

The team presents the first large vocabulary speech neuroprosthesis. For this purpose, they implanted a patient suffering from ALS with 2 microelectrode arrays each in 6v and area 44. Due to the disease, which was already diagnosed 8 years ago, the patient is indeed not able to produce intelligible speech. This is nicely demonstrated in two high-quality videos. With a total of 26 data recording sessions in different tasks, the researchers are able to show that neurons in 6v are nicely tuned to orofacial movements. The team then employs an RNN and a 3-gram language model in a Viterbi search to decode attempted speech (both audible and silent) with small and very large vocabularies.

This study presents a huge leap in the field of speech neuroprosthesis and will be of large interest to the wider scientific readership. It combines amazing data with sophisticated analysis and thereby produces results far above the current state of the art. To me, the most exciting highlights are:

- * First large vocabulary speech neuroprosthesis in a patient that is indeed anarthric. The communication speed and naturalness thereby exceed anything presented previously.

- * First implantation of Utah Arrays with the specific goal of creating a speech neuroprosthesis. This by itself is a huge feat. Interestingly, area 44 which is implicated in all theories of speech production did not provide a lot of useful information in this setup.

- * The neural decoding model from audible speech transferred almost perfectly to silently mouthed speech, highlighting how similar the underlying neural processes are despite the absence of laryngeal movements.

In addition to these highlights, the authors analyze several design aspects of speech neuroprostheses that can be of relevance for future (clinical) studies.

Overall, this article is very well written and presents a wonderful advance to the field. The authors write clearly and concisely without overclaiming and compare their results to the most relevant state of the art. Their conclusions are robustly underpinned by data and careful analysis. Naturally, they also appropriately credit the relevant previous work. I particularly enjoyed the description of the methods, which is easy to follow and does not try to obscure the procedure. Of course, it would be even better to share data and code, something I strongly suggest before publication of this work.

We are gratified that the reviewer feels that this paper is a huge leap forward that will be of broad interest, and we thank the reviewer for their thorough read of the manuscript and helpful suggestions.

Some comments and suggestions remain before this article can be accepted however:

Larger points:

(1) I am not convinced by the extrapolation of decoding results depending on the amount of electrodes in Fig. 4(b). The data in the graph sampling results depending on the number of used electrodes is of interest and important for future speech neuroprostheses. It is also clear to see that results have not saturated, yet. However, the extrapolation to even more electrodes makes a large number of assumptions which might not be true. Can we really implant 10 times as many electrodes and still get more useful information? If we have electrodes of the same size, we might end up in areas that do not provide meaningful information anymore. If we use smaller electrodes, they might all measure the same signal. So I believe it would be good to show the data without extrapolation.

Thank you for pointing out these limitations, we have now removed the extrapolation. Fig. 4b now appears as follows:

We now express more caution about extrapolating these results when introducing them in the Results section:

Next, we investigated how accuracy improved as a function of the number of electrodes used for RNN decoding. Accuracy improved monotonically with a log-linear trend (Fig. 4b – doubling the electrode count appears to cut the error rate nearly in half). ~~suggesting This suggests that an increased channel count~~ intracortical devices capable of recording from more electrodes (e.g., denser or more extensive microelectrode arrays) should may lead be able to achieve to higher-improved accuracies in the future, although the extent to which this downward trend will continue remains to be seen. (doubling the electrode count appears to cut the error rate approximately in half).

We are also more cautious in the Discussion section:

First, word error rate decreases as more channels are added, suggesting that intracortical technologies that record more channels should-may enable lower word error rates in the future.

(2) For the same plot, it would be good to include standard deviation or standard errors across the 100 seeds for each amount of channels to indicate the variability. Same applies to Figure S4(i).

Thank you for this suggestion, we have now added standard deviations to Fig. 4(b) and Figure S4(i), which appear as follows:

(3) The analysis for the preserved articulatory representation is not convincing to me (even though I fully believe that this is the case). The authors state that they do not have articulatory labels, as “we do not have ground truth knowledge of when each phoneme is being spoken” (line 146) and thus use saliency maps instead. They then choose some arbitrarily chosen EMA data of a single subject to compare against. I think this is not a very convincing analysis, as it purely analyzes what the neural network has learned (to best discriminate), as opposed to the actual neural data and compares it to some very arbitrary data. It would be much more straight-forward to simply use the trained RNN to provide phonemic labels on the data (which the authors describe how to do). Then one can easily use the IPA table to get the articulator movements during each point of time. This way, the authors would have a large sample of neural data for each articulator during attempted speech production and could do convincing statistics

that the articulators are indeed still represented. This would be a lot more convincing than a cosine similarity of one vector per phoneme.

Thank you for these suggestions on how to improve the analyses. As we understand it, the reviewer is raising the following distinct concerns:

(1) Using a single subject for the EMA data

(2) Using EMA data instead of the IPA table to quantify articulatory structure in phonemes

(3) Using what the decoder has learned, as opposed to analyzing the neural data directly combined with decoder-derived labels

(4) Analyzing aggregate structure instead of performing a statistical analysis across individual samples of neural data

We have added a new supplemental figure to directly address these four points. To address (1), we now compare the neural data to 12 subjects of EMA data combined across two publicly available datasets: the USC-TIMIT database [1] and the Haskins Production Rate Comparison database [2]. We show that the neural data is significantly correlated to all 12 subjects, with good agreement across 11 (one appears to be an outlier).

[1] Narayanan, S. *et al.* Real-time magnetic resonance imaging and electromagnetic articulography database for speech production research (TC). *The Journal of the Acoustical Society of America* **136**, 1307–1311 (2014).

[2] Tiede, M. *et al.* Quantifying kinematic aspects of reduction in a contrasting rate production task. *The Journal of the Acoustical Society of America* **141**, 3580–3580 (2017).

To address (2), we now additionally compare the neural data to an IPA-derived “place” code (consonants are grouped into separate factors depending on their place of articulation in the IPA table) and vowel formants (since the first and second formant of each vowel are generally accepted to be a robust quantification of how that vowel sounds and how it is produced). The neural data also shows good agreement with these alternative ways of quantifying consonant and vowel articulation. Nevertheless, we believe that EMA data provides a more unbiased and quantitative way to estimate articulatory structure in phonemes. Since phonemes are produced by the continuous kinematic positioning of articulators, quantifying this structure directly should be more accurate than conceptually classifying the phonemes into discrete articulation categories.

To address (3), we have now implemented the reviewer’s suggestion of using the decoder outputs to identify time windows of data belonging to each phoneme. For each phoneme, we averaged the neural features across all time windows belonging to that phoneme to generate a phoneme-specific neural representation vector. This process is not guaranteed to produce high quality results, since decoders trained with the CTC loss function are not constrained to output peak probabilities coincident to when the phoneme actually occurs in the data (e.g., they may delay their output if it is advantageous to do so). Additionally, the decoder may make labeling errors since it is an imperfect classifier, adding noise to the quantifications. Nevertheless, we found that when we trained bidirectional decoders (as opposed to causal decoders, which appear to delay their output), we achieved results that were highly consistent with the decoder saliency vectors. We now include this method in the supplemental figure.

Finally, we include a third method of quantifying the neural representation of phonemes: we used data from the single-phoneme speaking task (shown in Fig. 1). While this has the disadvantage of not quantifying phoneme representations during natural sentence production, and has limited data points (it only represents a single day of data collection), it is completely free of any decoder. This method also has good agreement with the EMA data and IPA table, again confirming the robust presence of articulatory structure in the neural representation of phonemes.

Regarding (4), we believe the shuffle control we are currently using, which computes the distribution of correlations between neural and articulatory vectors that would be expected by chance, should be a sound way to assess whether there is significant articulatory structure in the neural activity. Shuffling across single vectors, and not individual samples of data, assess statistical significance at the structural level – i.e., not whether the individual vectors themselves are non-zero or appropriately measured with enough samples, but whether random vectors of the same type could be expected to show the same level of articulatory structure. We think this is the more challenging and relevant statistical level. Additionally, neural vectors are computed across all days of data (consisting of 10k+ sentences), which should yield a statistically robust result.

Nevertheless, we appreciate the desire to see how robust the articulatory structure is across multiple independent samples. We therefore split the data across 6 independent folds, and computed the neural representations from only those folds, using the decoder labeling method suggested by the reviewer. This resulted in 6 extremely similar neural representations, confirming that it is a robust result.

SFig 8. Different methods of quantifying articulatory structure in neural activity and articulator kinematics all yield similar results. (a) Representational similarity across consonants (top) and vowels (bottom) for different quantifications of the neural activity (“Saliency Vectors”, “Decoder Labels”, and “Single Phoneme Task”) and articulator kinematics

(“Haskins M01”, “USC-Timit M1”, “Place + Formants”). Each square in a matrix represents pairwise similarity for two phonemes (as measured by cosine angle between the neural or articulatory vectors). Consonants are ordered by place of articulation (but with approximants grouped separately) and vowels are ordered by articulatory similarity (as measured by “USC-TIMIT M1”). These orderings reveal block-diagonal structure in the neural data that is also reflected in articulatory data. “Haskins M01” and “USC-Timit M1” refer to subjects M01 and M1 in the Haskins and USC-Timit datasets. “Place + Formants” refers to coding consonants by place of articulation and to representing vowels using their two formant frequencies. See methods for more details about how each representation was derived. (b) Correlations between the neural representations and the articulator representations (each panel corresponds to one method of computing the neural representation, while each column corresponds to one EMA subject or the place/formants method). Orange dots show the correlation value (Pearson r), and blue distributions show the null distribution computed with a shuffle control (10,000 repetitions). In all cases, the true correlation lies outside the null distribution ($p < 1e-4$). Correlation values were computed between consonants and vowels separately and then averaged together to produce a single value. (c) Representational similarity matrices computed using the “Decoder labels” method on 6 different independent folds of the neural data. Very similar representations across folds indicates that the representations are statistically robust (average correlation across folds = 0.79).

(4) I am a bit surprised that the Phoneme Error Rates in Table 1 are the same for the online results and the improved LM. The better language model should also improve the alignment of the phonemes and if the phoneme errors are exactly the same, how can it construct better words from it (as the WER is lower)? It would be great if the authors could check again.

We now make clear the phoneme error rates are reported on raw RNN output (before a language model is applied). Thus, those entries in the table should be the same. Thank you for raising this lack of clarity. The legend for table 1 now reads:

Table 1. Mean phoneme and word error rates (with 95% CIs) for the speech BCI across all evaluation days.

Phoneme error rates quantify the quality of the RNN decoder’s output before a language model is applied, while word error rates quantify the quality of the combined RNN and language model pipeline. Confidence intervals (CIs) were computed with the bootstrap percentile method (resampling over trials 10,000 times). “Online” refers to what was decoded in real-time, while “offline” refers to a post-hoc analysis of the data using an improved language model (“Improved LM”) or different partitioning of training and testing data (“Proximal Test Set”). In the proximal test set, training sentences occur much closer in time to testing sentences, mitigating the effect of within-day neural nonstationarities.

We also added clarification to the Results text:

Encouragingly, the RNN often decoded sensible sequences of phonemes before a language model was applied (Fig 2c). Phoneme error rates computed on the raw RNN output were 19.7% for vocal speech (20.9% for silent; see Table 1) and phoneme decoding errors followed a pattern related to speech articulation, where phonemes that are articulated similarly were more likely to be confused by the RNN decoder (SFig 5).

Smaller:

(1) On p.3 line 75, the authors write “These probabilities were then processed by a language model to ...”. This is not described accurately: The probabilities are combined with the language model in the Viterbi search, the language model itself just supplies probabilities for sequences of words and does not process probabilities of the RNN.

Thank you for that correction, the sentence now reads as follows:

These probabilities were then processed by combined with a language model to infer the most likely underlying sequence of words, given both the phoneme probabilities and the statistics of the English language (Fig 2a).

Suggestions:

(1) I believe that for future BCI studies and eventually even clinical trials, it is far more important to know how much training data needs to be collected than how much data could be used for adaptation from

each day. This is particularly true as no adaptation is the more realistic use case (as the authors write in line 470). So for me, it would be better to have S4(i) in the main manuscript instead of 4(c).

Thank you for this suggestion. We are hopeful that data from other participants may help in reducing the amount of total data required for any new person, similar to how speech recognition systems, when trained on many speakers, can adapt to new speakers quickly. Thus, we think that adaptation to nonstationarities is currently the most pressing problem (although both problems are very important, and cross-participant transfer remains to be verified in future work with additional participants). We have chosen to keep the original panel, although we appreciate the suggestion.

(2) For the section headings within the result section, I would prefer it if the main finding could be summarized instead of the analysis step. So instead of "Representation of orofacial movement", I would prefer "Orofacial movements are represented in 6v but not in area 44". Instead of "Decoding Attempted Speech", I would prefer "Attempted Speech can be decoded even when silently mouthed". Again, this is just a suggestion.

Thank you for this suggestion, unfortunately Nature subheadings are limited to 40 characters.

Referee #3 (Remarks to the Author):

I briefly respond to the items listed on the review page of nature, and list comments below

A Summary of the key results

results are rather overstated regarding relevance to BCI end users

B Originality and significance: if not novel, please include reference work is highly original

C Data & methodology: validity of approach, quality of data, quality of presentation more than adequate

D Appropriate use of statistics and treatment of uncertainties adequate

E Conclusions: robustness, validity, reliability

more attention is required to address the limitations regarding generalizability, recalibrations and longevity of functionality

F Suggested improvements: experiments, data for possible revision see comments below

G References: appropriate credit to previous work? adequate

H Clarity and context: lucidity of abstract/summary, appropriateness of abstract, introduction and conclusions

Generally good but amendments required to tone down some overstated claims

My Review:

Authors report on a single participant with 4 Utah arrays on pre central and inferior frontal left cortex. Words were decoded by assessing threshold crossing in 64 electrodes of one array in inferior pre central cortex, yielding 62 words/minute at an error rate of 9-23% depending on used vocabulary size. Data are reported for 8 days of testing

The patient could still produce some speech sounds albeit unintelligible, thus could still contract articular muscles.

Overall, the work is novel and indeed for the first time displays proof of concept of decoding speech for people with speech disabilities with Utah arrays.

We thank the reviewer for their thorough read of the manuscript and helpful suggestions, and are gratified that the reviewer feels that the results are novel and show a proof of concept for intracortical speech BCIs.

However, several issues dampen my enthusiasm. The paper would be considerably stronger if authors would take the comments into consideration and amend interpretation and most importantly conclusions

We would like to begin by saying that we appreciate the reviewer's points raised below concerning the limitations and generalizability of the current demonstration from a clinical point of view. We wholeheartedly agree and have done our best to make it clear that we see the work only as a "proof of principle" that high-performance neural decoding of attempted speech is possible using intracortical arrays. We do not see the work in its current form as a complete, clinically viable system. As the reviewer rightly points out, several longstanding obstacles remain for clinically translating intracortical systems: (1) longevity of the recording technology, (2) adaptation to signal nonstationarity without burdening the user with frequent retraining, and (3) a need for also testing the efficacy of these systems in people with complete locked-in syndrome. We have amended the discussion to explicitly raise these three issues, and to make it clearer that this study is a proof-of-concept demonstration - not a complete, clinically viable system:

Our demonstration is a proof of concept that decoding attempted speaking movements ~~from-with a large vocabulary intracortical recordings~~ is possible using neural spiking activity ~~a promising approach~~.

However, it is important to note that it does not, but it is not yet constitute a complete, clinically viable system. Work remains to be done to reduce the time needed to train the decoder and adapt to changes in neural activity that occur across days without requiring the user to pause and recalibrate the BCI (see ^{32,24-27} for initial promising approaches). Additionally, intracortical microelectrode array technology is still maturing ^{33,34}, and is expected to require further demonstrations of longevity and efficacy before widespread clinical adoption (although recent safety data is encouraging³⁵, and next-generation recording devices are under development^{36,37}). Furthermore, the decoding results shown here must be confirmed in additional participants, and their generalizability to people with more profound orofacial weakness remains an open question. Variability in brain anatomy is also a potential concern, and more work must be done to confirm that regions of precentral gyrus containing speech information can be reliably targeted.

While we ourselves are optimistic that these problems can and will be overcome, we do not attempt to do so here, and we have made our best attempt above to make this clearer for readers. Nevertheless, we believe that our study provides important motivation for the field to continue working on these problems, since it demonstrates several key principles which are required for such a clinical device to be feasible in the first place: successful large-vocabulary decoding, intermixed tuning within a small cortical area, and a preserved representation of speech in someone with paralysis. Again, we are very grateful to the reviewer for pointing out our miscommunication.

1) For one, multiple statements give readers the suggestion that the findings are of direct relevance for clinical use which any opinion is overstated given the limited evidence. Most importantly, since the participant can still move facial muscles involved in articulation (the patient makes obvious head movements following the rhythm of the words in a sentence) which contribute to sensory feedback and thus possibly to decoding (in the sensorimotor face area motor and sensory function is mixed in pre- and post-central gyrus), it is by no means clear whether the findings apply to people with full facial paralysis. In particular, the statement that 'Our demonstration is a proof of concept that decoding attempted speaking movements from intracortical recordings is a promising approach' is overly bold given that the participant was still capable of actual articulator/face movements. Even during silent speech actual movements were made ('mouthing the words') which means signals included sensory feedback plus actual motor output.

Thank you for pointing out this important limitation. We appreciate the concern about whether this result will generalize to people with full facial paralysis. Indeed, the only way to know for sure would be to replicate the work in another participant with more extensive orofacial weakness; as such, we now explicitly mention this in the discussion:

Furthermore, the decoding results shown here must be confirmed in additional participants, and their generalizability to people with more profound orofacial weakness remains an open question.

Additionally, we have reworded the following sentence so as not to come across as overreaching or overpromising:

Our demonstration is a proof of concept that decoding attempted speaking movements ~~from with a large vocabulary intracortical recordings~~ is possible using neural spiking activity a promising approach. However, it is important to note that it does not, but it is not yet constitute a complete, clinically viable system.

Nevertheless, we do believe it is relevant and valuable to show a working system in a person who retains some orofacial movement. In our experience, paralysis is often incomplete in some way. This is supported by a recent study of potential BCI users [1] that found that "incomplete" locked-in syndrome, which still prevented normal communication due to severe paralysis, was significantly more common than complete locked-in syndrome. Of those surveyed in [1], the majority had some remaining mouth/head/face movement.

[1] Pels, Elmar G.M., Erik J. Aarnoutse, Nick F. Ramsey, and Mariska J. Vansteensel. "Estimated Prevalence of the Target Population for Brain-Computer Interface Neurotechnology in the Netherlands."

Neurorehabilitation and Neural Repair 31, no. 7 (July 2017): 677–85.
<https://doi.org/10.1177/1545968317714577>.

Second, prior work on Utah array recordings in arm/hand area in people with more complete arm/hand paralysis has demonstrated a strong neural representation for attempted movement in the absence of overt movement (e.g., [2-5]). We are optimistic that a similar result will hold for orofacial area of motor cortex.

[2] Collinger, J. L. *et al.* High-performance neuroprosthetic control by an individual with tetraplegia. *The Lancet* **381**, 557–564 (2013).

[3] Simeral, J. D., Kim, S.-P., Black, M. J., Donoghue, J. P. & Hochberg, L. R. Neural control of cursor trajectory and click by a human with tetraplegia 1000 days after implant of an intracortical microelectrode array. *Journal of Neural Engineering* **8**, 025027 (2011).

[4] Willett, F. R. *et al.* Hand Knob Area of Premotor Cortex Represents the Whole Body in a Compositional Way. *Cell* (2020) doi:[10.1016/j.cell.2020.02.043](https://doi.org/10.1016/j.cell.2020.02.043).

[5] Hochberg, L. R. *et al.* Neuronal ensemble control of prosthetic devices by a human with tetraplegia. *Nature* **442**, 164–171 (2006).

2) Second, recalibration is required every day (involving several hundred sentences) which, unless convincingly solved, is a significant drawback for daily use.

Indeed, calibration time is an important issue that must also be addressed before clinical translation, and we mention it explicitly as a limitation in the discussion:

Work remains to be done to reduce the time needed to train the decoder and adapt to changes in neural activity that occur across days without requiring the user to pause and recalibrate the BCI (see ^{28–30} for initial promising approaches).

This is an area of active research, and we are optimistic that algorithms can be developed to automatically track the nonstationarities across time and recalibrate the system in the background without requiring the user to explicitly retrain the decoder [1-5]. For example, one new method uses an unsupervised approach to track a stable subspace of neural activity over time [3]. We see the current study as providing motivation for developing these algorithms.

[1] Jarosiewicz, B. *et al.* Virtual typing by people with tetraplegia using a self-calibrating intracortical brain-computer interface. *Science Translational Medicine* **7**, 313ra179-313ra179 (2015).

[2] Dyer, E. L. *et al.* A cryptography-based approach for movement decoding. *Nature Biomedical Engineering* **1**, 967–976 (2017).

[3] Degenhart, A. D. *et al.* Stabilization of a brain–computer interface via the alignment of low-dimensional spaces of neural activity. *Nature Biomedical Engineering* **4**, 672–685 (2020).

[4] Farshchian, A. *et al.* Adversarial Domain Adaptation for Stable Brain-Machine Interfaces. Preprint at <https://doi.org/10.48550/arXiv.1810.00045> (2019).

[5] Karpowicz, B. M. *et al.* Stabilizing brain-computer interfaces through alignment of latent dynamics. 2022.04.06.487388 Preprint at <https://doi.org/10.1101/2022.04.06.487388> (2022).

3) Third, longevity of Utah arrays function is a recurring source of concern for practical use in vulnerable end-users. Since Utah arrays have been reported to degrade within a few years, and since replacement requires resurgery and placement of new arrays in different locations due to scarring of the brain tissue, feasibility for longterm BCI use is not indicated sufficiently. Placing more arrays as suggested in the

discussion will only aggravate the problem of short longevity and renders less brain tissue to place new arrays. These concerns in my opinion should be regarded carefully, and claims should be toned down accordingly.

Thank you for raising this important point. Indeed, array longevity is a critical issue for intracortical BCIs. We now explicitly mention this in the Discussion:

Additionally, intracortical microelectrode array technology is still maturing^{33,34}, and is expected to require further demonstrations of longevity and efficacy before widespread clinical adoption (although recent safety data is encouraging³⁵, and next-generation recording devices are under development^{36,37}).

While no such study has yet been published, preliminary results from several studies indicate that arrays retain their functionality for several years in people, with multiple examples of retained functionality for 1000+ days [1-3]. Additionally, new devices are being developed which may retain their functionality better than Utah arrays [4-5]. Again, we see our work as providing motivation for designing improved devices. We simply see Utah arrays as helpful for enabling this line of research at the current time.

[1] Bullard, A. J., Hutchison, B. C., Lee, J., Chestek, C. A. & Patil, P. G. Estimating Risk for Future Intracranial, Fully Implanted, Modular Neuroprosthetic Systems: A Systematic Review of Hardware Complications in Clinical Deep Brain Stimulation and Experimental Human Intracortical Arrays. *Neuromodulation: Technology at the Neural Interface* **23**, 411–426 (2020).

[2] Willett, F. R., Avansino, D. T., Hochberg, L. R., Henderson, J. M. & Shenoy, K. V. High-performance brain-to-text communication via handwriting. *Nature* **593**, 249–254 (2021).

[3] Simeral, J. D., Kim, S.-P., Black, M. J., Donoghue, J. P. & Hochberg, L. R. Neural control of cursor trajectory and click by a human with tetraplegia 1000 days after implant of an intracortical microelectrode array. *Journal of Neural Engineering* **8**, 025027 (2011).

[4] Musk, E. & Neuralink. An Integrated Brain-Machine Interface Platform With Thousands of Channels. *J. Med. Internet Res.* **21**, e16194 (2019).

[5] Sahasrabudhe, K. *et al.* The Argo: A high channel count recording system for neural recording in vivo. *J. Neural Eng.* (2020) doi:10.1088/1741-2552/abd0ce.

4) Fourth, throughout the manuscript the performance is compared to previous papers repeatedly, notably a single paper by the Chang group. Given the speed at which performance is improved in general in speech decoding, such comparisons are not very informative and rather fleeting in nature. I strongly suggest to reduce such comparisons, since the paper is the first to show speech decoding with Utah arrays which should be sufficiently novel by itself without repeated comparisons to Chang's paper.

Thank you for this suggestion. We have reduced the number of comparisons to the NEJM Chang work, and re-worded the conclusion as to how intracortical approach differs from ECoG. We kept some comparisons in the paper, as we think it is important to situate the work in comparison to the literature, and the NEJM work is the most directly comparable and highest-performing alternative work.

The discussion now reads as follows:

People with neurological disorders such as brainstem stroke or amyotrophic lateral sclerosis (ALS) frequently face severe speech and motor impairment and, in some cases, completely lose the ability to speak (locked-in syndrome²⁹). Recently, BCIs based on hand movement activity have enabled typing speeds of 8-18 words per minute in people with paralysis^{8,30}. Speech BCIs have the potential to restore natural communication at a much faster rate, but have not yet achieved high accuracies on large vocabularies (i.e., unconstrained communication of any sentence the user may want to say)¹⁻⁷. Here, we demonstrated a speech BCI that can decode unconstrained sentences from a large vocabulary at a speed of 62 words per minute, using microelectrode arrays to record neural activity at single neuron resolution. To

~~our knowledge, this is Here, we demonstrated a speech BCI that can decode unconstrained sentences from a large vocabulary at a speed of 62 words per minute, the first time that a BCI has far substantially exceeded the communication rates that alternative technologies can provide for people with paralysis (e.g., eye tracking³¹). We were able to decode at high speeds with 2.7 times fewer errors than the prior state of the art for speech BCIs when evaluated on a matching 50-word vocabulary², made possible by using intracortical microelectrode arrays that record neural activity at single neuron resolution.~~

Additionally, we removed the direct comparison to the NEJM paper in Figure 2 and the Figure 2 legend.

(b) Word error rates (edit distances) are shown for two speaking modes (vocalized vs. silent) and vocabulary sizes (50 vs 125,000 words). Vertical lines indicate 95% CIs. ~~Word error rates are 2.7 times lower than the prior state of the art when using the same 50 word vocabulary (prior work indicated with a dashed black line, which should be compared to the blue line).~~

Finally, we removed the direct comparison to ECoG when mentioning the single-word decoding results (these are now mentioned in Fig. 1 without comparison to ECoG):

~~When applying a simple naïve Bayes classifier to this dataset, we could achieve a 95.5% accuracy (95% CI = [94.2, 96.7]), which outperforms prior work based on electrocorticographic recordings (47.1%²). This indicates that the performance advance demonstrated here is not due to the language model, but rather the increased decodability afforded by higher resolution intracortical recordings.~~

Nevertheless, we think it is important to compare to the state of the art, and to let the reader know how these performance numbers fit within the broader context of the field. Therefore, we leave two comparisons in the manuscript: one in the abstract, and one in the results section when describing the performance numbers.

Other Comments:

5) More detailed exploration of the neural sources contributing to high decoding performance would strengthen the paper. Is it correct to conclude that different phoneme and word representations are distributed across the face area? How do the results shown in Figure 1c-d can be interpreted taking into account the somatotopic representation of speech articulators on the sensorimotor cortex (see Bouchard et al. 2013)?

Thank you for this helpful suggestion. We have added new analyses and figures which we believe speak to these questions and strengthen the paper. Additionally, after examining the somatotopy and array differences more closely, we identified and corrected an indexing error in our code which generated the original array somatotopy maps. The corrected maps now reveal differences between the two 6v arrays (the ventral array now appears to be more speech-specific).

While the main result still holds (both arrays are tuned to both speech and orofacial movement, and both have highly intermixed somatotopy), the new results suggest that the upper and lower parts of 6v are not identical in function. This finding is consistent with resting state fMRI data from the Human Connectome Project [1] and from T12 that situates the ventral region of 6v as part of a language-related network. We now include this new resting state data in supplemental figure 2. Whether this result will hold in additional participants can only be answered in future work.

[1] Glasser, M. F. *et al.* A multi-modal parcellation of human cerebral cortex. *Nature* **536**, 171 (2016).

In Figure 1, we now report classification results by array (and by time) across orofacial movements, phonemes and words (Fig. 1e). These new results show interesting differences between the two 6v arrays; the ventral array appears to contain more information about speech, especially during the instructed delay / cue presentation period, while the dorsal array appears to contain more information about orofacial movements.

(e) Classification accuracy across time for each of the four arrays and three types of movement. Classification was performed with a 100 ms window of neural population activity beginning at each time point. Shaded regions show 95% confidence intervals. Gray lines show the time of speech onset (average sound envelope recorded by a microphone). Vertical dashed lines show the beginning of the delay and go periods. Horizontal dashed lines show chance classification performance.

These results are echoed in the corrected somatotopy maps in Fig. 1f, reproduced below:

(f) Tuning heatmaps for both arrays in area 6v, for each movement category. Circles are drawn if binned firing rates on that electrode are significantly different across the given set of conditions ($p < 1e-5$ assessed with a one-way ANOVA; bin width=800 ms). Shading indicates the fraction of variance accounted for by the across-condition differences in mean firing rate.

To quantify more directly how tuning to orofacial movements, phonemes and words are intermixed across and within arrays, we added the following new figure. It shows that many electrodes are tuned to multiple movement categories, and that both arrays contain electrodes tuned to every speech articulator and to phonemes/words.

Supplemental Figure 4. (a) Pie charts summarizing the number of electrodes that had statistically significant tuning to each possible number of movement categories (from 0 to 6), as assessed with a 1-way ANOVA ($p < 0.0001$). On the 6v arrays, many electrodes are tuned to more than one orofacial movement category (forehead, eyelids, jaw, lips, tongue, larynx). (b) Bar plots summarizing the number of tuned electrodes to each movement category and each array. The ventral 6v array contains more electrodes tuned to phonemes and words, while the dorsal 6v array contains more electrodes tuned to orofacial movement categories. Nevertheless, both 6v arrays contain electrodes tuned to all categories of movement.

Finally, we added a new supplemental figure to directly analyze which array contributed the most towards RNN decoding of speech, reproduced below:

Supplemental Figure 7. (a) Word error rates of a retrospective offline decoding analysis using just the ventral 6v array (left column), dorsal 6v array (middle column), or both arrays (right column). (b) Heatmaps depicting the (offline) increase in phoneme error rate when removing each electrode from the decoder by setting the values of its corresponding features to zero. Almost all electrodes seem to contribute to decoding performance, although the most informative electrodes are concentrated on the ventral array. The effect of removing any one electrode is small ($< 1\%$ increase in phoneme error rate), owing to the redundancy across electrodes.

This figure shows that while the ventral array alone outperforms the dorsal array alone, both arrays combined perform substantially better (32% WER from the ventral array decreases to 21% WER). The electrode contribution heatmaps show that a portion of electrodes on the ventral array clearly appear to contribute the most, although contributions come from both arrays and a wide range of electrodes.

In sum, these new analyses confirm that tuning to orofacial movements, phonemes and words appears highly intermixed within arrays and is present in both regions of area 6v, while highlighting a new interesting difference: the ventral region of 6v appears to be more specialized for speech, while the dorsal region appears more related to orofacial movement. These results show that, at a single neuron level, somatotopy appears more mixed than what was implied by Bouchard 2013.

Although we recognize that the differences between the two 6v arrays raise the possibility of many new neuroscience-focused analyses and experiments, we think that these figures strike a good balance between raising a salient point about array differences and keeping the cohesive focus of the paper on BCI decoding of speech.

6) After training did the participant need to calibrate the system by copying sentences every day?

Thank you for raising this lack of clarity. Yes, RNN training was performed at the beginning of each day. We reworded this paragraph in the Results section to make this point clearer:

To train the RNN, at the beginning of each day At the beginning of each RNN performance evaluation day, we first recorded training -data where T12 attempted to speak 260-480 sentences at her own pace (41 ± 3.7 minutes of data; -(sentences were chosen randomly from the switchboard corpus¹² of spoken English). A computer monitor cued T12 when to begin speaking and what sentence to speak. The RNN was then trained on this data in combination with all prior days' data, using custom machine learning methods adapted from modern speech recognition¹³⁻¹⁵ to achieve high performance on limited amounts of neural data. In particular, we used unique input layers for each day to account for across-day changes in the neural activity, and rolling feature adaptation to account for within-day changes (SFig 45 highlights the effect of these and other architecture choices). By the last day, our training dataset consisted of 10,850 total sentences. Data collection and RNN training lasted 140 minutes per day on average (including breaks).

Ideally, training time could be minimized or eliminated. We believe this is feasible, as offline decoding analyses with fewer (or no) training data still yielded comparable performance (shown in Fig. 4c, reproduced below).

7) Training of the RNN was done ' at the beginning of each day '. Does this refer to the 8 days of testing the decoders? Or does this refer to separate days prior to testing?

Thank you again for raising this lack of clarity. This refers to the 8 days of decoder evaluation – see above.

8) Authors report use of silence tokens, how were they determined? And was the participant instructed to pause between words?

Thank you for raising these questions. Silence tokens were automatically added to the end of each word, for all words. The participant was not instructed to pause between words. While we call it a “silence”

token, as this is traditional for speech recognition, we view it more generally as a “word separator” that allows the RNN to communicate information about how words should be separated to the language model.

We added the following clarification to the Figure 2 legend:

Phonemes are offset vertically for readability and “<sil>” indicates the “silence” token (which the RNN was trained to produce at the end of all words).

We also added the following explanation to the Methods section 3:

We automatically added a “silence” phoneme to the end of each word in order to denote the separation between words. We reasoned that this gives the RNN decoder the ability to communicate demarcations between words to the language model, and we found that it improved performance. Note that the silence token is not necessarily intended to model literal silence, as the participant may or may not be silent between each word (although she appeared to take brief pauses between many of the words, as can be seen in SVideo 1).

9) Why did the participant need to press a button to indicate end of sentence production?

We chose to use this setup to simplify the demonstration, as we were focused on demonstrating high-performance decoding of sentences. Automatic detection of sentence completion would be important for a clinically deployed system in a person with more complete paralysis, and should be the subject of future work.

10) In Fig 4 b: extrapolating the line is very speculative given it is not clear whether actual additional electrodes will yield additional discriminatory information. This claim should be presented with less confidence also in the discussion

Thank you for pointing this out, we have removed the extrapolation line in Fig. 4b. Nevertheless, given that word error rate is continuing to decrease as more electrodes are added without yet saturating, we think it is reasonable to conclude that some performance improvement could be expected by adding more electrodes (although future work is required to test this idea). The figure panel now appears as follows:

We now express more caution about extrapolating these results when introducing them in the Results section:

Next, we investigated how accuracy improved as a function of the number of electrodes used for RNN decoding. Accuracy improved monotonically with a log-linear trend (Fig. 4b – doubling the electrode count appears to cut the error rate nearly in half), ~~suggesting~~ This suggests that an increased channel count in intracortical devices capable of recording from more electrodes (e.g., denser or more extensive microelectrode arrays) should may lead be able to achieve to higher-improved accuracies in the future, although the extent to which this downward trend will continue remains to be seen. (doubling the electrode count appears to cut the error rate approximately in half).

We are also more speculative in the Discussion section:

First, word error rate decreases as more channels are added, suggesting that intracortical technologies that record more channels ~~should-may~~ enable lower word error rates in the future.

11) An improved offline decoding approach was reported that performed better when training and test sentence were closer together in time. Authors report that better models that adapt to signal non-stationarity should be developed. Yet how would that work in real time without requiring the participant to frequently copy text to update training (which would render the BCI system cumbersome at least)

Thank you for pointing this out. Developing algorithms that automatically adapt to non-stationarities (without requiring the user to pause to collect training data) is an active area of research [1-4]. We now make more explicit reference to this idea in the Results section:

These results indicate that substantial gains in performance are likely still possible with further language model improvements and more robust decoding algorithms that generalize better to nonstationary data (for example, unsupervised methods that track non-stationarities without requiring new training data²⁴⁻²⁷).

[1] Dyer, E. L. *et al.* A cryptography-based approach for movement decoding. *Nature Biomedical Engineering* **1**, 967–976 (2017).

[2] Degenhart, A. D. *et al.* Stabilization of a brain–computer interface via the alignment of low-dimensional spaces of neural activity. *Nature Biomedical Engineering* **4**, 672–685 (2020).

[3] Farshchian, A. *et al.* Adversarial Domain Adaptation for Stable Brain-Machine Interfaces. Preprint at <https://doi.org/10.48550/arXiv.1810.00045> (2019).

[4] Karpowicz, B. M. *et al.* Stabilizing brain-computer interfaces through alignment of latent dynamics. 2022.04.06.487388 Preprint at <https://doi.org/10.1101/2022.04.06.487388> (2022).

12) The authors state several times that they attribute the improvement in decoding performance to a much larger number of channels (page 6, lines 205-206, page 8 lines 248). It seems to me that the performance gains could rather be attributed to higher density or recordings and acquiring data at the single-neuron level. Can the authors clarify their position and address this more explicitly in the Discussion?

Thank you for raising this potential point of confusion. We were only intending to say that, since performance does not yet appear to have saturated as a function of the number of electrodes included in the decoder, higher channel count devices ought to be able to provide higher levels of performance in the future. We have reworded the Results section to make this clearer:

Next, we investigated how accuracy improved as a function of the number of electrodes used for RNN decoding. Accuracy improved monotonically with a log-linear trend (Fig. 4b ~~– doubling the electrode count appears to cut the error rate nearly in half~~), ~~suggesting~~ This suggests that an increased-channel intracortical devices capable of recording from more electrodes (e.g., denser or more extensive microelectrode arrays) should-may lead-be able to achieve to higher-improved accuracies in the future, although the extent to which this downward trend will continue remains to be seen. ~~(doubling the electrode count appears to cut the error rate approximately in half).~~

Figures

13) Support fig 1: legend does not explain what each figure represents. It is mentioned that each 8x8 matrix is one electrode but it seems to represent 64 electrodes on a single array. Moreover, mention is made of 128 electrodes without indicating they mean area 6v

Thank you for pointing out this lack of clarity. We have amended the legend as follows:

Example spike waveforms detected during a 10-s time window are plotted for each electrode (data were recorded on post-implant day 119). Each 8x8 grid corresponds to a single 64-electrode array, and each rectangular panel in the grid corresponds to a single electrode. ~~and each blue trace is a~~ Blue traces show

~~single-example~~ spike waveforms (2.1-ms duration). Neural activity was band-pass filtered (250-5000 Hz) with an acausal, 4th order Butterworth filter. Spiking events were detected using a -4.5 root mean square (RMS) threshold, thereby excluding almost all background activity. Electrodes with a mean threshold crossing rate of at least 2 Hz were considered to have 'spiking activity' and are outlined in red (note that this is a conservative estimate that is meant to include only spiking activity that could be from single neurons, as opposed to multiunit 'hash'). The results show that many electrodes record large spiking waveforms that are well above the noise floor (the y axis of each panel spans 580 μ V, whereas the background activity has an average RMS value of only 30.8 μ V). ~~On this day, In area 6v,~~ 118 electrodes out of 128 had a threshold crossing rate of at least 2 Hz ~~on this particular day in area 6v~~ (113 electrodes out of 128 in area 44).

14) To improve readability, I suggest that the authors add titles to individual plots on Figure 4S and axis labels to Figure 5S.

Thank you for these suggestions, we have amended the figures as indicated.

Referee #4 (Remarks to the Author):

Review of Willett et al. "A high-performance speech neuroprosthesis"

Summary of Key Results

This study reports on the performance of an intracortical microelectrode based speech-to-text BCI in a single participant with ALS. In the setting of a clinical trial protocol with IDE, the team implanted four 64-channel microelectrode arrays: two on "Broca's area" and two on orofacial motor cortex. Using state of the art modeling approaches, the team reports a 9.1% WER rate with a 50 word vocab and 23.8% WER with a 125,500 word vocab with speech decoded at 62 words per minute. The novel use of small high density microelectrode arrays shows spatially intermixed tuning to speech articulators, which provides evidence that speech BCIs can probably perform with less cortical coverage than previously described. I agree with the authors that the results demonstrate early feasibility of microelectrode-based BCIs and will pave the way for future clinical studies in larger groups of patients. The results are exciting.

Originality and Significance

This is a highly original first-of-a-kind study that uses microelectrode recordings from pars triangularis and orofacial motor cortex as a speech BCI. The major significance of the work is that it demonstrates early feasibility of this approach for BCI design. As the authors point out, the microelectrode recordings provide new information about the representation of orofacial movements at the single neuron level.

We are gratified that the reviewer feels the results are exciting, original and demonstrate early feasibility. We thank the reviewer for their thorough read of the manuscript and their insightful and helpful suggestions.

Data and Methodology

Standard implantation approach based on fMRI data used to lateralize language given that patient is historically L handed. 4 microelectrode arrays placed. Each training session was numbered in the methods.

Can authors specify how many hours were spent on each training session? This would be interesting, if not helpful, to really understand how much training these BCIs require of patients.

Thank you for this helpful suggestion. We now clarify both the duration of the training data itself (41 minutes on average) as well as the total data collection and RNN training time, including breaks (140 minutes on average).

To train the RNN, at the beginning of each day we recorded data where T12 attempted to speak 260-480 sentences at her own pace (41 ± 3.7 minutes of data). ~~(sentences Sentences~~ were chosen randomly from the switchboard corpus¹² of spoken English). A computer monitor cued T12 when to begin speaking and what sentence to speak. The RNN was trained on this data in combination with all prior days' data, using custom machine learning methods adapted from modern speech recognition¹³⁻¹⁵ to achieve high performance on limited amounts of neural data. In particular, we used unique input layers for each day to account for across-day changes in the neural activity, and rolling feature adaptation to account for within-day changes (SFig 45 highlights the effect of these and other architecture choices). By the last day, our training dataset consisted of 10,850 total sentences. Data collection and RNN training lasted 140 minutes per day on average (including breaks).

Ideally, training time would be minimized or eliminated for practical use in a clinical system. We believe this is feasible, as offline decoding analyses with fewer (or no) data still yielded comparable performance (shown in Fig. 4c, reproduced below).

Appropriate Use of Statistics and Treatment of Uncertainties
Analytic approaches seem appropriate to me.

Conclusions

The authors conclude that this work suggests that a higher channel count system that records from a small area 6v is a feasible path forward that will restore conversational speeds to people with paralysis. This is a reasonable conclusion of the work, though as the authors point out earlier, there is much more work to be done. I think it is appropriate that the authors temper their conclusions about how this approach compare's with Chang's ECoG-based approach (which they cite from NEJM) given that both approaches could yield clinically-valuable results in future studies.

Thank you for this suggestion. We have reduced the number of comparisons to the NEJM Chang work, and re-worded the conclusion as to how intracortical approach differs from ECoG. We kept some comparisons in the paper, as we think it is important to situate the work in comparison to the literature, and the NEJM work is the most directly comparable and highest-performing alternative work.

The discussion now reads as follows:

People with neurological disorders such as brainstem stroke or amyotrophic lateral sclerosis (ALS) frequently face severe speech and motor impairment and, in some cases, completely lose the ability to speak (locked-in syndrome²⁹). Recently, BCIs based on hand movement activity have enabled typing speeds of 8-18 words per minute in people with paralysis^{8,30}. Speech BCIs have the potential to restore natural communication at a much faster rate, but have not yet achieved high accuracies on large vocabularies (i.e., unconstrained communication of any sentence the user may want to say)¹⁻⁷. Here, we demonstrated a speech BCI that can decode unconstrained sentences from a large vocabulary at a speed of 62 words per minute, using microelectrode arrays to record neural activity at single neuron resolution. To our knowledge, this is ~~Here, we demonstrated a speech BCI that can decode unconstrained sentences from a large vocabulary at a speed of 62 words per minute,~~ the first time that a BCI has far-substantially exceeded the communication rates that alternative technologies can provide for people with paralysis (e.g., eye tracking³¹). ~~We were able to decode at high speeds with 2.7 times fewer errors than the prior state of the art for speech BCIs when evaluated on a matching 50-word vocabulary², made possible by using intracortical microelectrode arrays that record neural activity at single neuron resolution.~~

Additionally, we removed the direct comparison to the NEJM paper in Figure 2 and the Figure 2 legend.

(b) Word error rates (edit distances) are shown for two speaking modes (vocalized vs. silent) and vocabulary sizes (50 vs 125,000 words). Vertical lines indicate 95% CIs. ~~Word error rates are 2.7 times lower than the prior state of the art when using the same 50 word vocabulary (prior work indicated with a dashed black line, which should be compared to the blue line).~~

Finally, we removed the direct comparison to ECoG when mentioning the single-word decoding results (these are now mentioned in Fig. 1 without comparison to ECoG):

~~When applying a simple naïve Bayes classifier to this dataset, we could achieve a 95.5% accuracy (95% CI = [94.2, 96.7]), which outperforms prior work based on electrocorticographic recordings (47.1%²). This indicates that the performance advance demonstrated here is not due to the language model, but rather the increased decodability afforded by higher resolution intracortical recordings.~~

Nevertheless, we think it is important to compare to the start of the art, and to let the reader know how these performance numbers fit within the broader context of the field. Therefore, we leave two comparisons in the manuscript: one in the abstract, and one in the results section when describing the performance numbers.

Suggested Improvements

Pg 2. Lines 45-46: I think you mean to say “ventral motor cortex” not ventral premotor cortex.

We mean to refer to area 6v as “premotor cortex”, in contrast to area 4 (primary).

Pg 2. Second paragraph: it is unclear here what task is being performed to generate phonemes. Is this the same as the orofacial movement task described in the prior paragraph? I think so based on reading the methods, but some clarity would be helpful in the main text.

Thank you for pointing out this lack of clarity. It is indeed the same task, which we now clarify in the main text with the following addition:

To investigate this, we recorded neural activity from four microelectrode arrays, two in area 6v⁹ (ventral premotor cortex) and two in area 44 (part of Broca’s area), while our BrainGate study participant attempted to make individual orofacial movements, speaking single phonemes, or speaking single words in response to cues displayed on a computer monitor (Fig. 1a-b; SFig 1 shows recorded spike waveforms).

Pg3. 118 – 119 – I agree with the advantages of intracortical MER, however, since there are so many other factors that influence decodability, I’m not sure the authors should make blanket statements about this being responsible for their increased decodability result.

Thank you for pointing this out. We do not want to overreach in our conclusions, so we have removed this statement.

I would make it clear in the main text exactly how much training (in hours) to perform each of the decoding tasks.

Thank you again for this helpful suggestion, which we addressed above.

References

References seems restricted to a small number of groups consider citing more work by Kai Miller, Nathan Crone, Eddie Chang...etc.

Thank you for this suggestion. We want to cite broadly and give appropriate credit to the field. We have now added the following references:

Flinker, A. *et al.* Redefining the role of Broca’s area in speech. *Proceedings of the National Academy of Sciences* **112**, 2871–2875 (2015).

Kellis, S. *et al.* Decoding spoken words using local field potentials recorded from the cortical surface. *Journal of neural engineering* **7**, 056007 (2010).

Pei, X., Barbour, D. L., Leuthardt, E. C. & Schalk, G. Decoding vowels and consonants in spoken and imagined words using electrocorticographic signals in humans. *J. Neural Eng.* **8**, 046028 (2011).

Andrews, J. P. *et al.* Dissociation of Broca's area from Broca's aphasia in patients undergoing neurosurgical resections. *Journal of Neurosurgery* **138**, 847–857 (2022).

Kanas VG, Mporas I, Benz HL, Sgarbas KN, Bezerianos A, Crone NE. Joint spatial-spectral feature space clustering for speech activity detection from ECoG signals.

Moses DA, Mesgarani N, Leonard MK, Chang EF. Neural speech recognition: continuous phoneme decoding using spatiotemporal representations of human cortical activity

Reviewer Reports on the First Revision:

Referees' comments:

Referee #1 (Remarks to the Author):

The authors have submitted a revised manuscript. The revised manuscript includes interesting additional somatotopy results (previously obscured by an indexing error) in Figure 1, differentiating orofacial movements from phonemes. The results are echoed in corrected somatotopy maps. Supplemental Figure 7 shows how this somatotopy contributes to contributions by individual electrodes. In supplemental Figure 2 it is interesting that the hotspot on the individual scan corresponds to the ventral 6V array. Additional revisions in response to reviewer comments add further clarity. Certain queries, which admittedly would involve considerable additional analyses, have been artfully and eloquently deferred to future work. In summary, the authors have addressed all major concerns and I congratulate them on their fine work.

Referee #2 (Remarks to the Author):

The authors have addressed all of my concerns. I am particularly impressed by the thorough additional analyses carried out for the articulator representations I asked for.

Even though they didn't incorporate all of my suggestions, they gave convincing reasons when they did not.

It is also great to read that the authors have decided to make code and data available to the public! In my opinion this truly makes a difference.

I am looking forward to seeing this article published,
Christian Herff

Referee #3 (Remarks to the Author):

I thank the authors for their thoughtful and detailed responses to my comments. Added new analyses and figures clarify the study and results quite well. Modified language regarding implications for end-users satisfies my concerns. I have no further comments on this well-conducted and -presented study.

Nick Ramsey

Referee #4 (Remarks to the Author):

The authors have performed a comprehensive revision that responds to each of my queries.